# A compendium of DNA-binding specificities of transcription factors in *Pseudomonas syringae*

Ligang Fan [1,2,4], Tingting Wang[2,4], Canfeng Hua[2,4], Wenju Sun [1,4], Xiaoyu Li[1], Lucas Grunwald [2], Jingui Liu[2], Nan Wu[1], Xiaolong Shao[2], Yimeng Yin[3], Jian Yan [1,2✉] & Xin Deng [2✉]

*Pseudomonas syringae* is a Gram-negative and model pathogenic bacterium that causes plant diseases worldwide. Here, we set out to identify binding motifs for all 301 annotated transcription factors (TFs) of *P. syringae* using HT-SELEX. We successfully identify binding motifs for 100 TFs. We map functional interactions between the TFs and their targets in virulence-associated pathways, and validate many of these interactions and functions using additional methods such as ChIP-seq, electrophoretic mobility shift assay (EMSA), RT-qPCR, and reporter assays. Our work identifies 25 virulence-associated master regulators, 14 of which had not been characterized as TFs before.

[1] School of Medicine, Northwest University, 710069 Xi'an, China. [2] Department of Biomedical Sciences, City University of Hong Kong, Kowloon Tong, Hong Kong, SAR, China. [3] Translational Research Institute of Brain and Brain-Like Intelligence, Shanghai Fourth People's Hospital Affiliated to Tongji University School of Medicine, 200081 Shanghai, China. [4] These authors contributed equally: Ligang Fan, Tingting Wang, Canfeng Hua, Wenju Sun. ✉email: jian.yan@cityu.edu.hk; xindeng@cityu.edu.hk

Transcription factors (TFs) play a central role in transcriptional regulation mediating various biological events[1,2]. The regulatory function of TF is primarily exerted via its binding to genomic DNA in a sequence-specific manner. Such action directly interprets the genome function, taking the first step in decoding the portion that is involved in the *cis*-regulation of transcription[3]. Therefore, information of the DNA-binding specificities of TFs is required to further dissect the mechanism of this regulatory code. To date, extensive efforts have been made to systematically illustrate the binding motifs of the vast majority of TFs in a variety of eukaryotic species, including human[4], mouse[5], fruit fly[6], nematode[7], and yeast[8,9]. However, a similar scale of data remains generally lagging for prokaryotic microorganisms.

*Pseudomonas syringae* is a Gram-negative and model pathogenic bacterium, causing devastating plant diseases and economic loss worldwide[10,11]. In *P. syringae*, essential biological pathways, including plant virulence, are known to be under the control of a large group of TFs[12–20]. For example, HrpS is a major TF of the needle-like type III secretion system (T3SS) that directly secretes a group of effector proteins into hosts, causing serious diseases[12]. HrpS forms a heterohexamer with HrpR to activate the expression of *hrpL*, which is also regulated by the sigma factor RpoN[13,14]. HrpL is an alternative sigma factor and can activate the expression of T3SS genes by binding to the motif 5′-GGAAC-N16-17-CCACNNA-3′ in the *hrp* promoters[21]. Through genetic screening, a variety of TFs have been found positively or negatively regulating T3SS genes and mediating bacterial pathogenicity, among them, including AefR[16,17], RhpR[18,19], and CvsR[20], revealing the importance of studying TFs in understanding the pathogenesis and finding potential targets for the cure. Unfortunately, due to lack of data systematically revealing the TF-binding specificities, the transcriptional regulators governing the important biological processes in *P. syringae* remains largely underexplored.

In order to globally profile the DNA-binding specificities of all TFs in *P. syringae*, we have purified proteins for all 301 annotated TFs in the genome[22] and applied a high-throughput systematic evolution of ligands by exponential enrichment (HT-SELEX) approach[4]. Analysis of the data reveals binding specificities of 100 distinct TFs described by 118 different position weight matrix (PWM) motifs, including 106 (90%) showing dimer-binding specificity, 8 (7%) with monomer-binding, and 4 (3%) with trimer-binding modes, respectively. Both biological and technical replicative data from the DNA-binding domain and full-length protein of the same TF reveal the same binding motif. In addition, ChIP-seq and biochemical validations are carried out to further confirm the consistency of our data, demonstrating the overall high quality of the results. To further identify the genome-wide interactions between these TFs and their target loci in the genome, we have mapped an intricate binding network of 100 TFs, assigning 25 TFs targeting genes related to virulence. These results reveal a group of master TFs in multiple key pathways, including the known RhpR in T3SS[18,19] and a few master TFs in other virulent pathways, such as c-di-GMP, bacterial motility, surface attachment, siderophore, and reactive oxygen species (ROS). The results provide not only a global view of binding preference and functional analyses of TFs in *P. syringae,* but also a valuable resource to support future studies in *Pseudomonas* and other related pathogens.

## Results

### HT-SELEX revealed binding specificities of 100 TFs in *P. syringae*.
To characterize the molecular basis of all the annotated TFs in *P. syringae* in terms of DNA binding, we performed a previously established HT-SELEX pipeline[4]. In this study, we used the model pathogen strain *Pseudomonas savastanoi* pv. *phaseolicola* 1448A (previously known as *Pseudomonas syringae* pv. *phaseolicola* 1448A), a member of the *P. syringae* species group. A set of coding sequences of both DNA-binding domains (DBDs) and full-length *P. syringae* TFs were compiled based on annotations in the DBD: Transcription-factor prediction database (www.transcriptionfactor.org)[22], altogether resulting in 301 TFs included in the present study (Fig. 1a). Almost all analyzed TFs contained the DNA-binding domains (DBDs) of helix–turn–helix structure (HTH)[23,24], and they could be further classified into 31 families partially grouped by relevant function[25,26]. The majority of TFs (163 TFs) belonged to five families, including LysR family, TetR family, GntR family, OmpR family, and AraC/XylS family (Supplementary Table 1a). We successfully cloned 301 full-length TFs and 9 DBDs into the pET28a vector and then expressed their corresponding 6×His-tagged proteins in *E. coli* BL21 cells (Fig. 1a and Supplementary Table 1b). After four rounds of HT-SELEX on all TFs starting with a DNA pool containing 40-bp double-stranded DNA with randomized sequences adapted to parallel sequencing systems (Supplementary Fig. 1 and Supplementary Table 1c), robust enrichment of specific DNA sequence was obtained for 100 TFs (Fig. 1a). The DNA-binding specificity was described as a PWM[4] (Supplementary Data 4). Of the top five largest TF families, the GntR and OmpR families reached the highest success rate of 50%, while the AraC family had a low success rate of 5%. In summary, HT-SELEX generated DNA-binding motifs for TFs that belonged to 26 different families[27] (Supplementary Table 1a).

We then re-categorized these TFs into 69 distinct modules, according to the similarity of PWMs (Fig. 1b and Supplementary Data 1). The results showed that for most modules, the TFs and motifs showed a one-to-one relationship, which is consistent with the notion that the paralogous TF genes were evolved through genome rearrangement and horizontal duplication[28]. Different from the multicellular organisms that contain many cell types, the pattern of gene expression in a prokaryotic cell is relatively simple and may only require different TFs to recognize the same binding site occasionally under ad hoc circumstances. Therefore, only in a few modules, more than one TF share similar PWMs, although we witnessed a cluster of nine TFs displaying similar DNA-binding specificities (Fig. 1b and Supplementary Data 1).

**Most TFs bound to homodimeric sites**. Analysis of the PWM width revealed the length of motifs ranging from 6 bp up to 27 bp in *P. syringae*, with the most common width sitting around 16 bp (Fig. 2a). Compared to the mouse and human TFs whose monomeric binding sites are usually 10 bp long or even shorter[4,5], the length of binding sites for most TFs in *P. syringae* exceeded such a scale, indicating that they were likely to prefer dimeric sites. We further classified their corresponding PWMs into three types, including monomer, dimer, and multimer. The results showed that 91% (107 motifs) of PWMs were indeed of dimeric site type, while <7% (8 motifs) and 3% (3 motifs) of PWMs belonged to the monomer or trimer-binding specificity, respectively (Fig. 2b).

The dimeric binding type could be further separated into three subtypes given different monomer orientations, including head-to-head, head-to-tail, and tail-to-tail. We defined the binding mode that two monomers of one TF bound in opposite strands of the DNA forming a homodimer as the head-to-head orientation. If two directions of such opposite homodimers were observed, we defined the other one as the tail-to-tail orientation (for detailed PWM information, please see Module 18 in Supplementary Data 1). Our result showed that 85% (91 motifs) of PWMs displayed a head-to-head palindromic binding site, while 14%

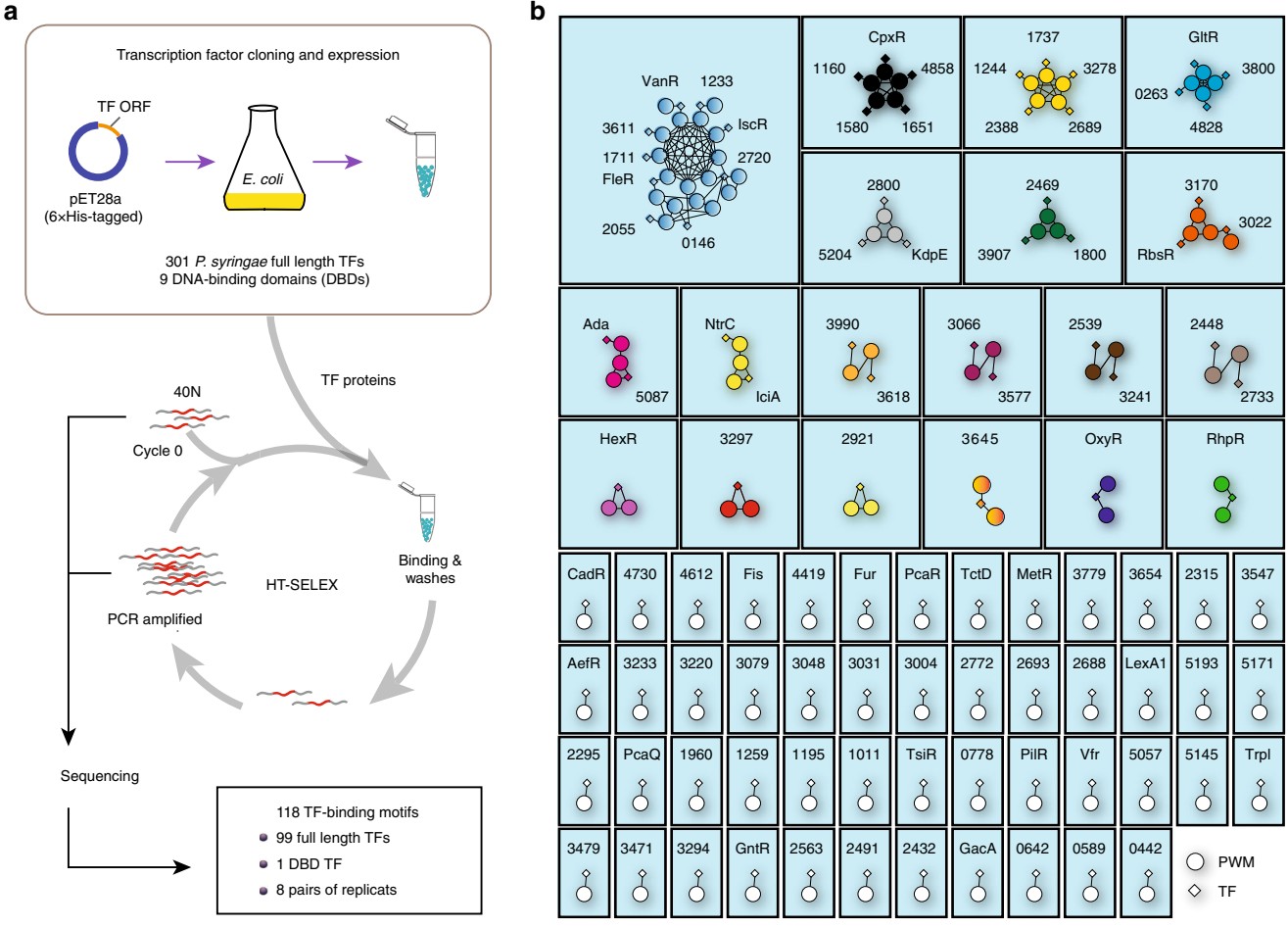

**Fig. 1 Network representation of the similarity of the PWMs from the HT-SELEX assay. a** Schematic description of protein expression and high-throughput SELEX process. **b** Network of similarity of the obtained PWMs. Diamonds indicate TF genes, and circles indicate individual PWMs. Edges are drawn between a TF and its PWM models, and between similar PWM models (SSTAT similarity score >E-05). The TFs without names are named with their locus tag omitting PSPPH_. TF transcription factor, PWM position weight matrix. See also Supplementary Fig. 1, Supplementary Data 1, Supplementary Data 3, and Supplementary Table 1.

(15 motifs) of PWMs exhibited a head-to-tail orientation and only one motif showed a tail-to-tail mode among 107 dimeric site type PWMs, indicating the predominant orientation preference for head-to-head homodimeric binding (Fig. 2b). In addition, such preference was observed for almost all TF families (Supplementary Fig. 2a).

To verify the reliability of the binding motifs and the biological function of TF putative binding sites in vitro and in vivo, a series of experimental validations were performed. PcaQ regulates the expression of genes relevant to the degradation of the aromatic acid protocatechuate (3,4-dihydroxybenzoate) by binding to the promoter of its target *pcaH* operon[29]. We showed that PcaQ displayed a monomeric binding preference with a core consensus motif of GGTTATG. Using its PWM motif, 33 putative binding sites were predicted in the genome (defined as PWM-predicted binding site with $P <$ E-05 hereafter, Supplementary Data 2), including sites located in the promoters of *pcaH*, PSPPH_0985, and *katB*. We first cloned the genomic fragments carrying these sites and applied the electrophoretic mobility shift assays (EMSA, see "Methods"). The results confirmed all three interactions (Fig. 2c and Supplementary Fig. 2b). We then constructed a *pcaQ* gene deletion strain (Supplementary Fig. 2c), and expected that the absence of PcaQ would result in dysregulation of its target genes. Interestingly, the expression levels of both PSPPH_0985 and *katB* were significantly

upregulated, implying that PcaQ was a transcriptional repressor that bound to their promoters.

By contrast, MetR, a known TF involved in oxidative stress tolerance and pathogenicity[30], displayed a head-to-head binding model (Fig. 2d). The PWM of MetR was then applied to scan the entire genome, resulting in 47 putative target sequences ($P <$ E-05 and Supplementary Data 2). We chose three of them for validation with EMSA, including the promoters of PSPPH_2621 (an esterified fatty acid cis/trans isomerase), PSPPH_1024 (a homocysteine S-methyltransferase), and PSPPH_4247 (a GAF domain/GGDEF domain/EAL domain-containing protein), respectively. As expected, all these sites were bound by MetR, but not for the negative control region (Fig. 2d and Supplementary Fig. 2d). Further experiments were carried out to test the biological function of the MetR binding in vivo. We reason that if the validated MetR-binding sites (defined as putative sites that were also verified by experimental methods, such as EMSA or ChIP, hereafter) were functional, the deletion of *metR* in cells should lead to the disrupted expression of its target genes. In line with the hypothesis, we first checked the expression of the target gene PSPPH_2621, whose promoter was bound by MetR (Fig. 2d). We confirmed that the transcription of PSPPH_2621 was significantly downregulated in a strain deficient of MetR (Δ*metR*) compared with that in the wild-type strain (Fig. 2d and Supplementary Fig. 2e). Subsequently, to avoid the indirect

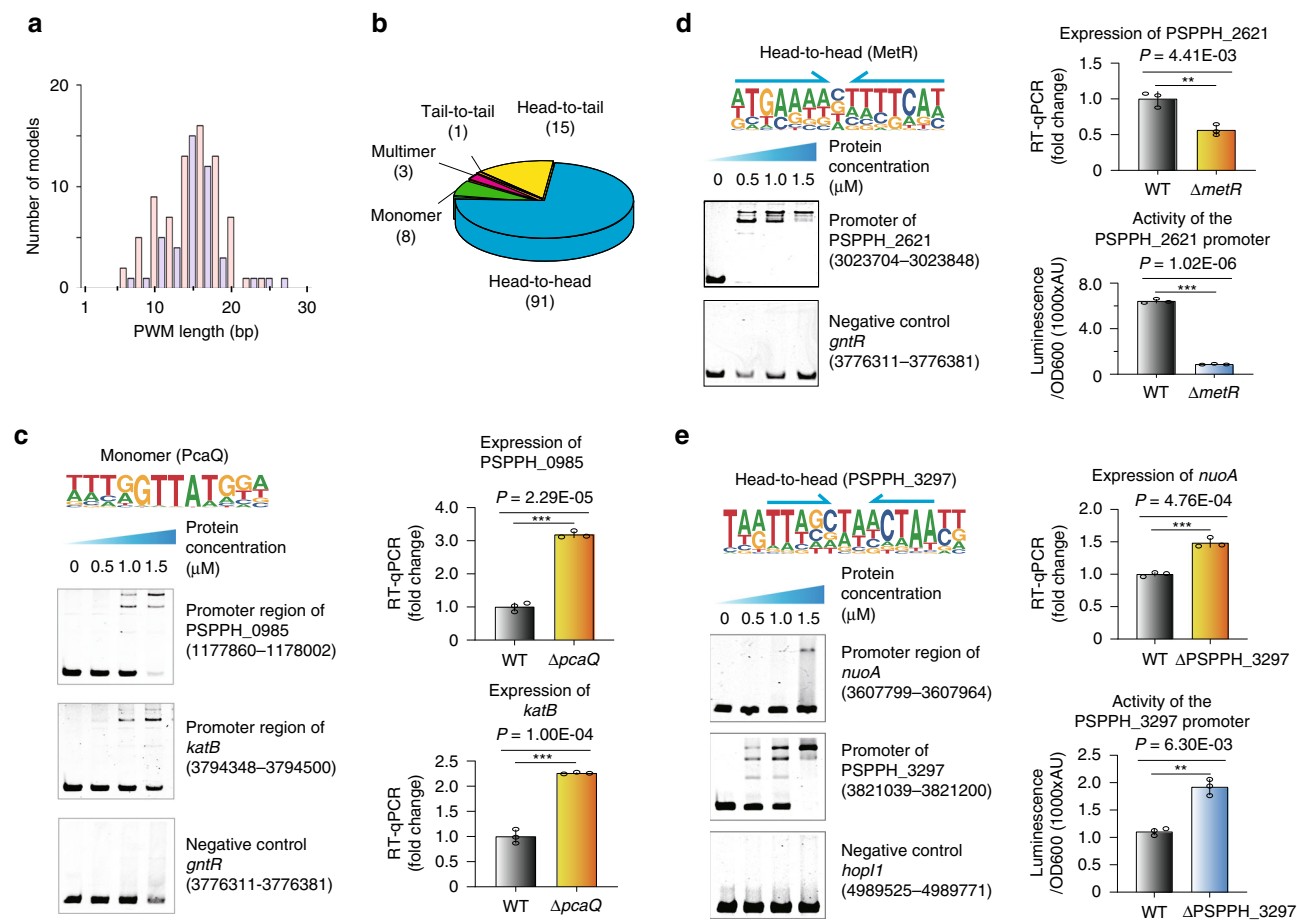

**Fig. 2 Validation of different TF-binding modes. a** Histogram shows the distribution of PWM model lengths. Note that TFs prefer even (red) over odd (blue) lengths due to more dimeric motifs, and that the specificity of most PWM length shows ~15 bp. **b** Pie chart shows the category of different TF-binding modes. Classification of all binding models into non-repetitive sites (monomer), and sites with two or three similar subsequences (dimer and multimer, respectively). The dimeric types are further classified as head-to-head, head-to-tail, and tail-to-tail. **c** Validation of the binding of PcaQ by EMSA and RT-qPCR. The putative binding sites from the promoter of target gene PSPPH_0985 and *katB* were verified using EMSA, respectively, and a fragment taken from the *gntR* promoter was used as the negative control. The expression of target gene PSPPH_0985 and *katB* was measured in wild-type and *pcaQ* mutant (*ΔpcaQ*) by RT-qPCR, respectively. Logo shows the binding site model of PcaQ. **d** Validation of the binding of MetR by EMSA, RT-qPCR, and a reporter assay. The binding site from the promoter of target gene PSPPH_2621 was verified using the EMSA, and a fragment taken from the *gntR* promoter was used as the negative control. The expression of target gene PSPPH_2621 was measured in wild-type and *metR* mutant (*ΔmetR*) by RT-qPCR. The promoter activity of target gene PSPPH_2621 was evaluated in wild-type and *metR* mutant by the reporter assay. Logo shows the binding site model of MetR, with its monomeric half-sites indicated with arrows. *P* value is 4.410E-03 (upper). *P* value is 1.017E-06 (lower). **e** Validation of the binding site of PSPPH_3297 by EMSA, RT-qPCR, and a reporter assay. The putative binding sites from the promoter of target gene *nuoA* and PSPPH_3297 were verified using the EMSA, respectively, and a fragment taken from the *hopI1* promoter was used as the negative control. The expression of target gene *nuoA* was measured in wild-type and PSPPH_3297 mutant (*ΔPSPPH_3297*) by RT-qPCR. The promoter activity of target gene PSPPH_3297 was evaluated in wild-type and PSPPH_3297 mutant by a reporter assay. Logo shows the binding site model of PSPPH_3297, with its monomeric half-sites indicated with arrows. *P* value is 4.760E-04 (upper). *P* value is 6.300E-03 (lower). Statistic *P* values by two-tailed Student's *t* test are shown; *$P < 0.05$, ***$P < 0.001$ (**c–e**). Three independent biological replicates were performed (**c–e**). Error bars show standard deviations. TF transcription factor, PWM position weight matrix. Also see Supplementary Figs. 2, 3, and Supplementary Data 4.

interference, we tested the biological importance of this binding ex vivo with a reporter assay, in which we cloned the PSPPH_2621 promoter region carrying the MetR validated binding site immediately upstream of a promoter-less luciferase gene in a pMS402-*lux* reporter plasmid. The result showed a significantly lower level of luciferase activity in the Δ*metR* strain than the wild-type cells, indicating the positive regulation of MetR on PSPPH_2621 transcription (Fig. 2d). These findings demonstrated that MetR directly bound to the PSPPH_2621 promoter and activated its expression.

Another example, PSPPH_3297, a MarR family transcriptional regulator, had a compact dimer-binding preference in a head-to-head orientation with the monomeric half-site beginning with a TT dinucleotide (Fig. 2e). By searching for the genome-wide PSPPH_3297 putative binding sites with this PWM model, 29 target loci were identified ($P < $ E-05; Supplementary Data 2). We verified the binding preference in vitro and in vivo. The PSPPH_3297 protein was predicted to bind two consecutive sites in its own promoter with the proximal site spanning from 65 bp to 48 bp upstream of ATG (chr: 3821136-3821153) and the distal site starting from 83 bp to 66 bp upstream of ATG (chr: 3821154-3821171). Accordingly, we amplified a 162-bp DNA fragment containing the two putative binding sites and validated the binding of PSPPH_3297 with EMSA (Fig. 2e). Since both validated sites are within its own promoter, it is infeasible to directly examine the impact of PSPPH_3297 deletion on the

target gene expression in vivo. Alternatively, we referred to the reporter assay to scrutinize the biological role of PSPPH_3297 binding to these sites. Interestingly, we found that the promoter activity was significantly higher in a mutant strain of PSPPH_3297 (ΔPSPPH_3297) compared with the wild-type strain (Fig. 2e and Supplementary Fig. 2f). These results suggested that PSPPH_3297 acted as a transcriptional repressor when binding to its own promoter, playing a negative auto-regulatory role, which had consistently reproduced the previously well-described phenomenon in *E. coli* for decades[31].

Besides these examples, we carried out similar validation experiments for a few additional TFs, including RbsR, PSPPH_4419, PSPPH_1651, TrpI, and PSPPH_3170 (Supplementary Fig. 3), and the results showed that almost all sites predicted by our PWM models could be recovered. To this end, we have demonstrated that HT-SELEX generated the TF-binding motifs of high quality, which could well predict the TF putative binding sites in the *P. syringae* genome.

**Systematic mapping of TF targets revealed primary T3SS regulators in *P. syringae*.** To comprehensively delineate the interactions between TFs and their target genes, we scanned the *P. syringae* genome using all PWMs generated in this study (Supplementary Data 2). Based on our cutoff ($P <$ E-05), the number of putative genomic binding sites for a TF ranged from 6 (PSPPH_2563) to 1481 (PSPPH_3577) with a median of 70 (Fig. 3a). To check whether the differential number of putative genomic binding sites for different TFs resulted from the motif degeneracy, we plotted the number of sites in the genome against the information content for all motifs. No significant correlation was observed between the two variables, suggesting that the distribution of target genes per TF was unlikely caused by the motif degeneracy itself but could be attributable to the *P. syringae* genome composition, likely driven by evolution (Supplementary Fig. 4a).

Since the pathogenicity of *P. syringae* largely depends on T3SS[32], we were particularly interested in recognizing the TFs that were primarily involved in regulating the T3SS genes. The T3SS component genes are clustered in a 25-kb pathogenicity-related island, whereas the majority of the effector genes are dispersed in the genome[33]. By specifically connecting TFs and target genes that were involved in T3SS, the resulting network included 45 TFs and 46 T3SS genes (Fig. 3b). The targets implicated genes of all types of components and effectors in T3SS, covering multiple *hrp* and *hrc* genes for eliciting the hypersensitive response in non-host plants and causing pathogenesis in susceptible host plants, as well as a few *hop* and *avr* effector genes capable of triggering disease symptoms[34,35]. In summary, our results unveiled that 40 TFs had multiple targets, and 25 T3SS genes were under the control of more than one TFs, suggesting a highly intricate regulatory network underlying the pathogenicity system of *P. syringae*.

Master regulators are commonly known in the development of a multicellular organism as a subset of TFs that appears to control the expression of multiple downstream genes and govern the lineage commitment[36]. Here, we employed the term "master regulator" to specify a class of functionally crucial TFs that participated in a pathway or a biological event by regulating multiple downstream genes associated with that event. In line with this definition, we identified TFs whose targets were significantly enriched in the T3SS genes (hypergeometric test, $P < 0.01$). Among these 45 TFs in the network, TrpI, RhpR, GacA, and PSPPH_3618 were shown to act as the master regulators in T3SS (Fig. 3c). RhpR and GacA have been well recognized as T3SS regulators in *P. syringae*[18,19,37–40]. RhpR is a repressor in

self-regulation and regulation of a cascade of T3SS genes to influence bacterial virulence, including *hrpR*, *hopR1*, *flhA*[18,19,37,38]. GacA was found to regulate T3SS by directly binding to the promoter and activating the expression of T3SS regulators HrpRS, and thus modulated the expression of T3SS cascade genes, including alternate sigma factors *hrpL*[39,40].

Our network showed that RhpR was an auto-regulatory TF that bound to its own promoter, consistent with our previous findings[18,19]. Meanwhile, it regulated transcription of eight T3SS genes, including both component genes (*hrcQa*, *hrpS*, and *hrpZ1*) and effector genes (*hopAA1*, *hopAT1*, *hopAJ1*, *hopAK1*, and *avrB2*) (Fig. 3b). Notably, a binding site (coordinates: 103671–103686 in the large plasmid) was predicted in the promoter of a T3SS effector gene *avrB2*, supported by the ChIP-seq data[37] in vivo (Supplementary Fig. 4b). We previously reported that RhpS was a repressor of RhpR, whose expression was elevated in the *rhpS* knockout strain (Δ*rhpS*) compared to the wild-type *P. syringae* (Supplementary Fig. 4c). By performing the de novo motif discovery analysis in the ChIP-peaks, a head-to-head homodimeric motif was obtained, similar to our HT-SELEX-generated motif but in degenerate quality (Supplementary Fig. 4d), probably due to the small number of peaks called in the ChIP-seq (103 significant peaks)[37]. Upon deletion of RhpS, the increased expression of RhpR consequently turned down the transcription of the *avrB2* gene, suggesting that RhpR directly bound to the promoter of *avrB2* and suppressed its transcription (Supplementary Fig. 4b).

To verify the biological function of the master TF, PSPPH_3618, which was still an uncharacterized gene, a series of biochemical and phenotypic experiments were performed (Fig. 3d–j). We first cloned the genomic fragments carrying the motif and performed EMSA to confirm that PSPPH_3618 indeed interacted with the promoters of both *rhpPC* operon and *avrB2* gene in vitro (Fig. 3d). Then we generated a bacterial strain deficient of PSPPH_3618 (ΔPSPPH_3618; Supplementary Fig. 4e), and inspected the transcription of its target genes (*avrB2* and *rhpP*) in both nutrient-rich (KB) and nutrient-deficient (MM) media, respectively. T3SS genes were expressed at a very low level in *P. syringae* when grown in KB, but could be induced to high levels in MM or in plants[41,42]. As a result, we showed that the transcriptional levels of *avrB2* and *rhpP* were both significantly lower in ΔPSPPH_3618 than in the wild-type strain in both media (Fig. 3e–h), supporting the PSPPH_3618-mediated transcriptional regulation of these two T3SS genes in vivo. We further infiltrated the wild-type and ΔPSPPH_3618 strains into the primary leaves of bean plants. After 6 days post-inoculation, we observed that the plant infected by ΔPSPPH_3618 strain displayed milder disease symptoms than by the wild-type *P. syringae* strain on the same leaves (Fig. 3i). To quantify the virulence, we extracted and counted the bacteria from the inoculated sites and found significantly over tenfold less bacterial growth of the mutant strains than the wild-type strain (Fig. 3j). To this end, we concluded that the putative master regulator PSPPH_3618 was functionally crucial in bacterial virulence, possibly through its implication in modulating T3SS genes, the *rhpPC* operon, and a T3SS effector AvrB2 known to enhance bacterial virulence[42,43].

**Master TFs in other virulence pathways.** Besides T3SS, the virulence of *P. syringae* is also controlled by other factors, such as c-di-GMP[44], flagella[34,45], ROS[44], toxins[46], siderophores[44], and surface attachment factors[47]. To gain a more comprehensive understanding of the regulation of virulence in *P. syringae*, we mapped the putative binding sites of all TFs for which we had obtained motifs to the genes that were involved in all of the

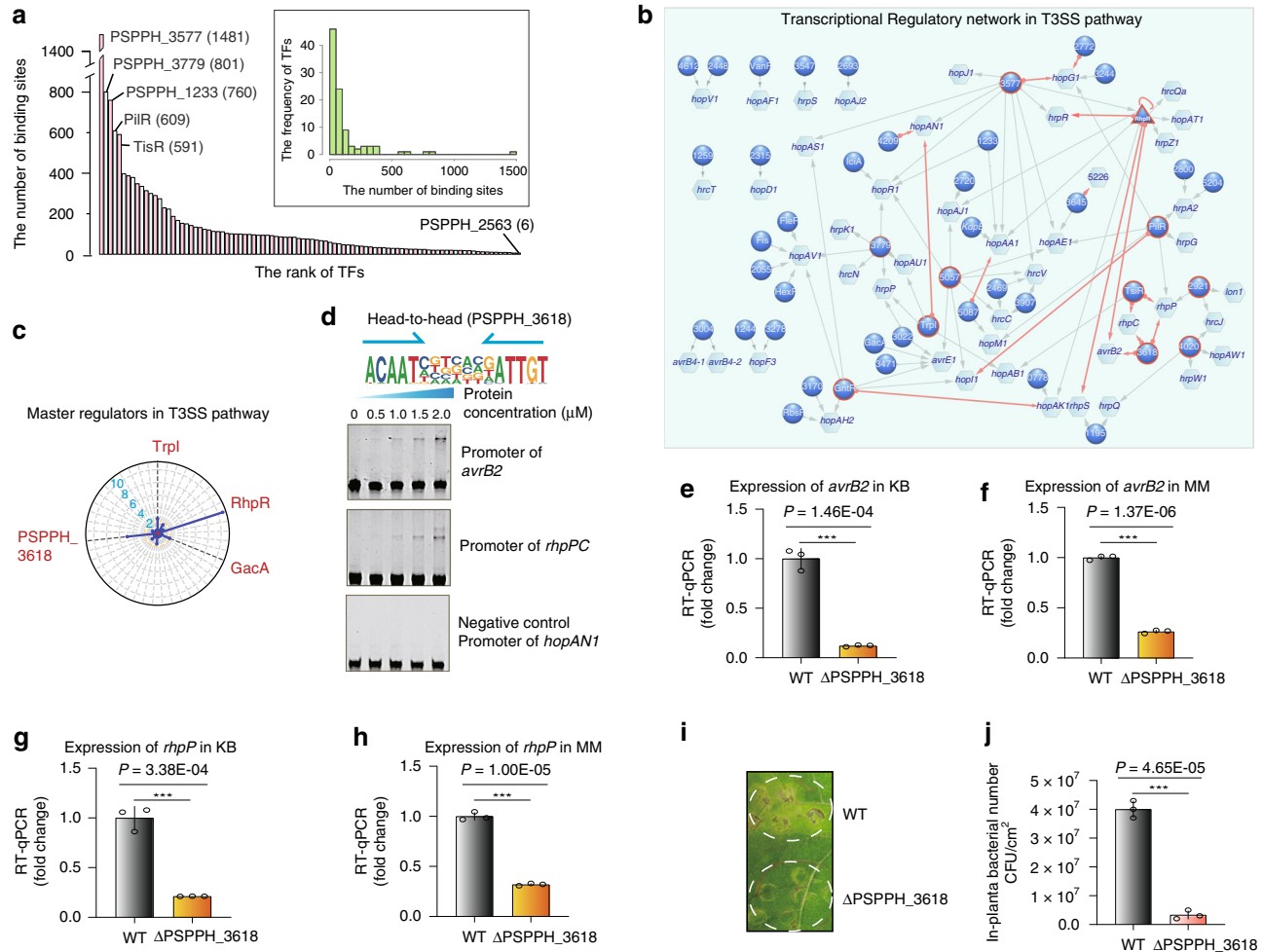

**Fig. 3 Transcriptional regulation in the T3SS pathway. a** Bar chart shows the distribution on the number of the putative binding sites per TF. Note that most of TFs target fewer than 200 genes with five TFs exceptional bind more than 500 sites in the genome. **b** Network illustrates the regulatory relationship between TFs and their target genes in the T3SS pathway. Circles indicate TF proteins, hexagons indicate target genes, and the triangle represents the TF with auto-regulatory activity. TFs that can regulate more than two target genes are highlighted by the red edges. Red arrows show the putative binding sites located in the promoters of the target genes, while gray arrows show the putative binding sites located in the target gene bodies. The TFs without names are named with their locus tag omitting PSPPH_. **c** Radar plot shows the putative master regulators identified in the T3SS pathway. Each radiation line represents a TF, and the radial length of thick colored line is −log10 (P value), representing the significance of the enrichment of the TF targets within T3SS-associated genes. The names of master regulators are marked in red-type face. **d** The head-to-head binding motif of PSPPH_3618 (upper) and the validation of the putative binding sites of PSPPH_3618 by EMSA (lower). The validated binding sites were from the promoters of the *rhpPC* operon and *avrB2*. The promoter of *hopAN1* was negative control. **e, f** The expression of target gene *avrB2* was measured in the WT and ΔPSPPH_3618 strains by RT-qPCR in KB and MM, respectively. P value is 1.460E-04 (**e**). P value is 1.373E-06 (**f**). **g, h** The expression of target gene *rhpP* was measured in the WT and ΔPSPPH_3618 strains by RT-qPCR in KB and MM, respectively. P value is 3.38E-04 (**g**). P value is 1.001E-05 (**h**). **i** Disease symptoms caused by either WT or ΔPSPPH_3618 strain, $10^5$ CFU/ml bacteria of which were infiltrated into the primary leaves of the bean. Disease symptoms were photographed 6 days after inoculation. **j** Bacterial numbers were quantified 6 days after inoculation. P value is 4.650E-05 (**j**). Three independent replicates were performed. Statistic P values by two-tailed Student's *t* test are shown. **$P < 0.01$ and ***$P < 0.001$ (**g, h, j**). Error bars show standard deviations. TF transcription factor.

aforementioned non-T3SS virulence pathways (Fig. 4 and Supplementary Data 3).

The secondary messenger c-di-GMP has the pleiotropic effects on flagella-mediated bacterial motility, surface attachment, siderophores, and ROS[44]. Our analysis identified a few putative master regulators participating in c-di-GMP, including MetR, PSPPH_2693, and PSPPH_1800 ($P < 0.01$) (Fig. 4a). To note, MetR was predicted to bind the promoter region of PSPPH_4247 that encodes a known c-di-GMP-specific phosphodiesterase. The binding was further confirmed by EMSA (Supplementary Fig. 2d). Interestingly, 13 TFs including two master regulators MetR and PSPPH_1800, co-bound to the promoter region of PSPPH_4420, which encodes a putative diguanylate cyclase[48], depicting a

sophisticated control over the cellular abundance of PSPPH_4420 (Supplementary Data 3a).

Flagella is an indispensable structure for bacterial swimming motility and contributes to the pathogenesis of bacteria through chemotaxis[34]. This study identified two highly confident motility-related master regulators: FleR and PSPPH_2720 (Fig. 4b). Motif-matched putative binding sites of FleR were found in the operon region of two important flagella genes, *fliE* and *fliH* (Supplementary Data 3b), which had been reported by a previous study[49]. FleR and PSPPH_2720 shared similar DNA sequence specificities (Fig. 1b), suggesting that they co-regulated the similar downstream targets (Supplementary Data 3b). The division of their respective role remains to be further explored, which could

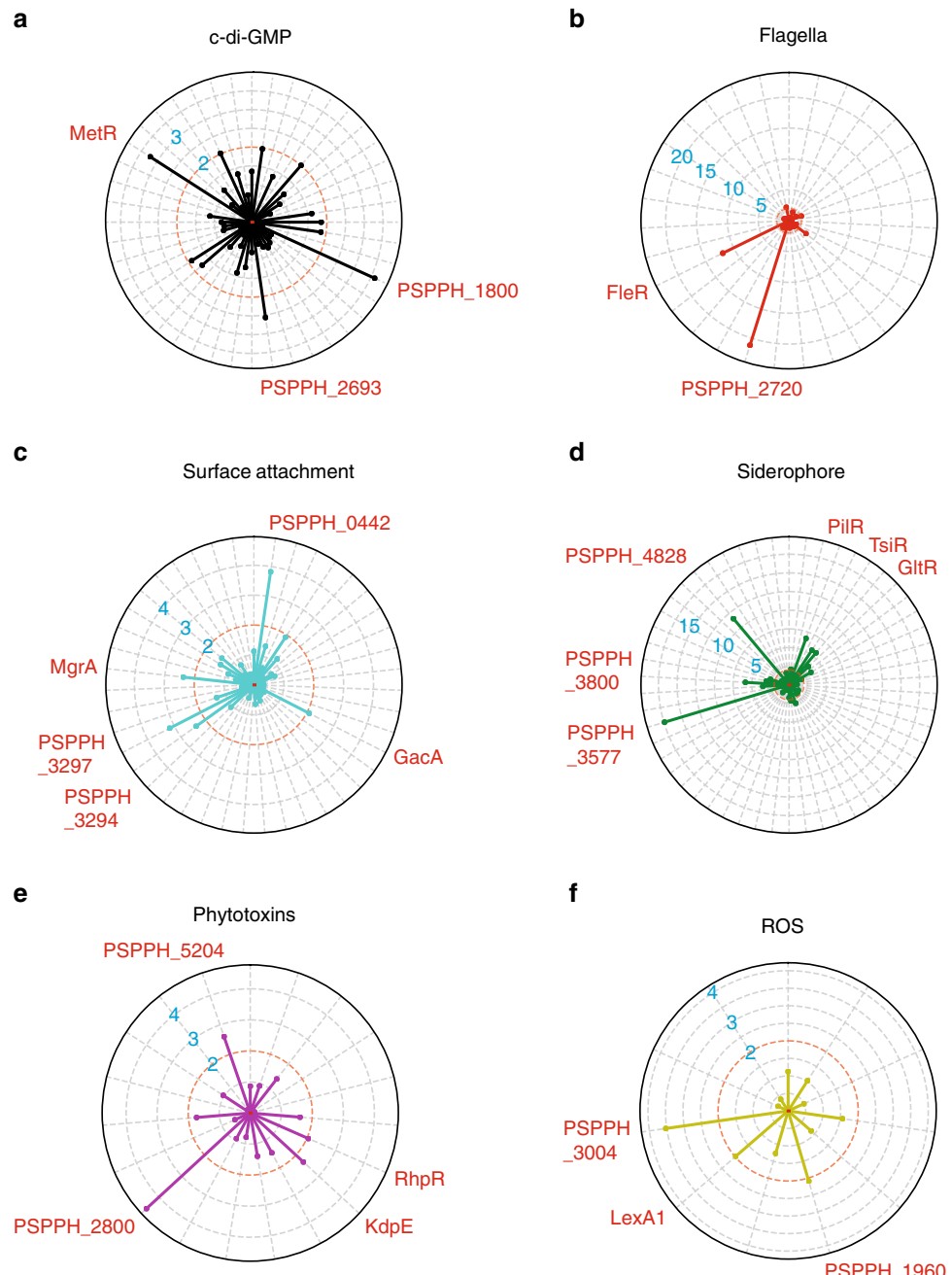

**Fig. 4 Master regulators in non-T3SS virulent pathways. a–f** Radar plots show the putative master regulators identified in six different non-T3SS virulent pathway, including c-di-GMP (**a**), flagella (**b**), surface attachment (**c**), siderophore (**d**), phytotoxins (**e**), and SOD (**f**). Each radiation line represents a master regulator, and the radial length of the thick colored line is −log10 (*P* value), representing the significance of the enrichment of the TF targets within each pathway. The names of master regulators are marked in red-type face. Also see Supplementary Data 3. TF transcription factor.

be either in a redundant mechanism or due to duty switching under different physiological conditions.

Furthermore, type IV pili are also required for virulence and chemotaxis, under the regulation of the surface attachment factors, such as PilM, PilN, PilO, PilP, PilQ, PilR[47]. In total, five TFs were tested as the master regulators in regulating the surface attachment (Fig. 4c). Among them, PSPPH_3645 (a homolog of ROS-sensing regulator MgrA in *Staphylococcus aureus*)[50] bound to the promoter regions of four genes (*mucA*, *mucB*, PSPPH_2653, and *algU*) within this pathway (Supplementary Data 3c). Meanwhile, we noted that a surface attachment factor, also being a transcription factor PilR, shared multiple targets with PSPPH_3645, including *mucA*, *mucB*, and *algU*, even though it

was not recognized as a master regulator in this category (Supplementary Data 3c). Interestingly, PilR turned out to be a master regulator of the siderophore pathway (Fig. 4d), governing the transcriptional regulation of as many as 18 genes serving iron transport (Supplementary Data 3d), inferring the interconnectivity of both pathways.

Microbial siderophores (iron chelators) contribute to iron dyshomeostasis in host plants through perturbation of heavy-metal homeostasis. In *P. syringae*, siderophore production is regulated by PvdS, PvdE, PvdP, and PvsA[44,51]. In this pathway, six TFs displayed strong involvement in regulating siderophore-relevant genes, recognized as the master regulators, including PilR, GltR, TsiR (PSPPH_0857), PSPPH_4828, PSPPH_3800, and

PSPPH_3577 (Fig. 4d). TsiR bound to the promoter region of the *fecBCDE* operon and regulated up to 14 genes in this functional pathway, suggesting an essential role of TsiR in mediating iron transport in *P. syringae* (Supplementary Data 3d).

Moreover, different pathovars of *P. syringae* produce a number of secondary metabolites called phytotoxins, causing chlorosis or necrosis in planta[46]. For example, *P. s.* pv. *phaseolicola* produces phaseolotoxin to cause halo blight on legumes[46]. Following our analyses, we found that four TFs have potential putative binding sites in toxins-related genes (Fig. 4e and Supplementary Data 3e). ROS are metabolic byproducts of aerobic respiration, which are responsible for maintaining redox homeostasis in cells as well as immunity of the host against some microorganisms[52]. Super-oxidase dismutases (SOD), such as SodA, SodB, KatG, and KatE, are required by bacteria to defend ROS[53]. We also reported that three TFs (LexA1, PSPPH_3004, and PSPPH_1960) were significantly participating in ROS resistance (Fig. 4f and Supplementary Data 3f). In sum, we showed 23 TFs acting as master regulators in a variety of non-T3SS virulence pathways. To our best knowledge, ten of them were previously annotated as TFs (MetR, FleR, LexA1, RhpR, KdpE, PilR, TsiR, GltR, GacA, and MgrA), while the other 13 of them remained uncharacterized by the time we initiated the study (PSPPH_3297, PSPPH_3294, PSPPH_1800, PSPPH_2693, PSPPH_2720, PSPPH_3004, PSPPH_1960, PSPPH_5204, PSPPH_2800, PSPPH_4828, PSPPH_3800, PSPPH_3577, and PSPPH_0442).

## Discussion

Genome-wide studies of transcriptional regulation in prokaryotic microorganisms are not common[54]. In this study, we conducted HT-SELEX to successfully profile the DNA-binding specificities of 100 distinct TFs with 118 PWM models in the model pathogen *P. syringae* by HT-SELEX, covering almost one-third of all putative TFs in its genome. We also performed replicative experiments for TFs without enrichment sequence, but consistently all of them still turned out unsuccessful in reaching a specific motif. Therefore, the failure of sequence enrichment of some TFs is likely owing to that they are not sequence specific TFs or that the sequence preference of them is generally too weak relative to other successful cases. Most of *P. syringae* TFs exhibited unique binding motifs yielding totally 69 distinct motif clusters, likely due to the small number of TFs and low demand for redundant regulation that is usually required under complex circumstances. Using PWMs to scan the genome of *P. syringae*, we obtained a global view of its transcriptional regulatory network. Special attention was paid to virulence pathways known to be involved in pathogenesis. Putative binding sites of some TFs showed strong enrichment in these pathways. For example, RhpR was identified as a master regulator in T3SS, consistent with previous work[18,19]. In addition to the T3SS pathway, we also identified master regulators for six different virulence-relevant pathways.

We asked whether the biochemically determined DNA-binding specificity of a TF could accurately reflect its genomic binding in vivo. We compared the HT-SELEX-generated motif with a PWM model produced by ChIP-seq in *P. syringae* cells[37] of the same TF for which both datasets were available. As expected, the ChIP-seq identified a similar head-to-head homodimeric-binding motif to the HT-SELEX generated model, although with lower or missed information content at some positions. For example, the A nucleotide in the tail position of a monomeric half-site was biochemically required, suggested by the HT-SELEX model, where the information content was almost 0 bit in the PWM model of ChIP-seq (Fig. 3d). This also suggests the suboptimal quality of PWM models from ChIP-seq, likely due to the limited number of

peaks in the small bacterial genome used to generate the model or the complex DNA-binding mode in vivo sometimes mediated by protein–protein interactions. However, we certainly appreciate that ChIP-seq has its own advantage as it detects TF in vivo binding sites that contain epigenetic features e.g., DNA modifications, the context of promoter site, including sigma-factor-binding sites, etc., which cannot be shown by in vitro studies. Therefore, these methods complement each other and provide different information on TF-DNA binding.

Yet another interesting finding was that most of the binding models by these prokaryotic TFs fell into the category of homodimers in a head-to-head orientation (90/118) with only very few (8/118) monomeric binding modes. By contrast, many human TFs tend to bind monomeric sites, inferring that the monomeric binding mode is evolutionarily preferred in multicellular eukaryotic organisms. The prevalence of monomeric binding sites in the genome is dramatically higher than that of the dimeric sites, which allows a higher incidence of juxtaposed binding by multiple different TFs for collaborative regulation of the target genes[55]. This is very important for tissue-specific gene regulation, but may not be so important for the unicellular prokaryotes. Crowded binding of many different TFs for collaborative regulation may even curb a swift response upon external stimuli, which are usually faced by bacteria exposed to the complicated living environment. As a matter of fact, the TF-binding clusters are less favored in prokaryotic microorganisms (Supplementary Data 2).

TFs regulate bacterial virulence by influencing both T3SS and non-T3SS pathways in *P. syringae*. This study revealed 25 TFs as potential virulence-associated master regulators. Interestingly, we observed interactions between these putative T3SS-associated master regulators and non-T3SS virulence pathways, as well as interactions among different non-T3SS pathways. For example, the binding consensus sequence of PilR was located in the promoter regions of genes associated with four different virulence pathways, including T3SS, surface attachment, c-di-GMP, and siderophore pathways. In addition, RhpR was found to bind the promoters of genes associated with both T3SS and flagella-mediated motility. Similar crosstalk was also detected for other TFs, such as TsiR, GntR, MgrA, FleR, KdpE, TrpI, MetR, PSPPH_2921, and PSPPH_3577. Remarkably, the metabolism-associated TFs TrpI (tryptophan synthesis)[56] and GntR (gluconate utilization)[57] exhibited direct regulation of bacterial virulence by regulating PSPPH_2701 (an alkylhydroperoxidase in ROS) and PSPPH_2571 (an insecticidal toxin complex protein in toxins) (Supplementary Data 3e and 3f), and HopAK1 (a T3SS helper protein) (Fig. 3b), respectively. Taken together, the results indicate intensive interconnectivity among different pathogenicity systems in *P. syringae*, and suggests possible crosstalk between metabolism and virulence.

Overall, our study provides a valuable resource for binding preferences of the vast majority of TFs in the *P. syringae* genome, which will facilitate future studies to disentangle the molecular mechanisms in various biological processes in *P. syringae* and other related organisms.

## Methods

**Bacterial strains, culture media, plasmids, and primers**. The bacterial strains, plasmids, and primers used in this study are listed in Supplementary Information. The *P. syringae* 1448A strain and its derivatives were grown at 28 °C in KB (King's B) medium with shaking at 220 rpm or on KB agar plates. The *E. coli* strains were grown at 37 °C in LB broth (Luria-Bertani) with shaking at 220 rpm or on LB agar plates. Antibiotics for *E. coli* and its derivatives were used at the following concentrations: for *E. coli* with pET28a, pMS402, and pK18mobsacB, kanamycin at 50 μg/ml; for *E. coli* with pHM1, spectinomycin at 50 μg/ml. Antibiotics were used for *P. syringae* and its derivatives at the following concentrations: mutants of *P. syringae*, rifampicin at 25 μg/ml; *P. syringae* with pHM1 plasmids for ChIP-seq,

rifampicin at 25 μg/ml, and spectinomycin at 100 μg/ml; *P. syringae* with pMS402 and plasmid for *lux* detection, rifampicin at 25 μg/ml and kanamycin at 100 μg/ml; *P. syringae* with pK18mobsacB for mutant construction, rifampicin at 25 μg/ml and kanamycin at 100 μg/ml.

**Cloning and recombinant protein purification.** Oligonucleotides and vectors used for cloning of His-tagged proteins in this study are listed in Supplementary Table 1. The 301 TFs were identified, according to Wilson et al.[22]. DNA fragments of 301 TFs were acquired from the *P. syringae* 1448A genome and amplified by polymerase chain reaction (PCR) to obtain the encoding regions of full-length or DNA-binding domain genes, respectively. Each forward or reverse PCR primer carried a 20-bp sequence identical to the linearized plasmid sequence at the 5′ end or 3′ end of the cutting site followed by the gene-specific sequence. Therefore, the homologous match of the two 20-bp recombination fragments determined the direction of the target gene in the expression vector. Then the *Bam*HI-linearized pET28a vector and individual TF PCR products (containing 20-bp overlapped sequences on 5′- and 3′ end, respectively) were mixed in the molar ratio of 1:2, and then incubated with recombinase (Vazyme ClonExpress II One Step Cloning Kit, Vazyme Biotech) for 30 min at 37 °C. The recombination products were chemically transformed into *E. coli* (DH5 α) competent cells with at 42 °C for 1 min. Finally, the successful constructs verified by *Bam*HI digestion were further transformed into the *E. coli* BL21 (DE3) strain for protein purification. Subsequently, a single colony on each plate was picked and inoculated into 3-ml-sterilized LB broth containing 50 μg/ml kanamycin for 12 h. Then, we transferred the culture into 300 ml sterilized LB broth and grown at 37 °C, 220 rpm to $OD_{600} = 0.6$. In all, 1 mM isopropyl β-D-1-thiogalactopyranoside (IPTG) was added into the bacterial culture to induce each TF expression at 16 °C for 16 h. The bacterial culture was then centrifuged at 4 °C, 5000 ×*g* for 5 min to harvest the pellet. The whole process was performed at 4 °C. The pellet was suspended in 12 ml buffer A (500 mM NaCl, 25 mM Tris-HCl, pH 7.4, 5% glycerol, 1 mM dithiothreitol, 1 mM phenylmethanesulfonyl fluoride (PMSF) and lysed by sonication at ten seconds interval for 20 min and then centrifuged at 4 °C (10,000 ×*g*, 30 min) to collect protein supernatant. The supernatant was filtered with a 0.45-μm filter, and the filtrate was added into a Ni-NTA column (Bio-Rad), which had been equilibrated with buffer A before using. The Ni-NTA column was eluted with 30-ml gradient from 60 to 500 mM imidazole prepared in buffer A gradually. Eluted fractions from 300 mM to 500 mM were pooled, and sodium dodecyl sulfate-polyacrylamide gel electrophoresis (SDS-PAGE) was used to verify the molecular weight of target TFs.

**HT-SELEX.** HT-SELEX was performed by using selection ligands containing an 8-bp barcode before and after the 40 bp randomized region for Illumina system or a 10 bp barcode after the 40 bp randomized region for BGI system, respectively. The ligand libraries were produced by PCR amplification using the primers in Supplementary Table 1c and 40-bp randomized oligo as a template. The libraries were sequenced using Illumina HiSeq Xten or BGI MGISEQ 2000 sequencer to analyze the uniformity of each base (A, T, C, G). Then, 100–200 ng protein and 5-μl selection ligands were added into Promega binding buffer (10 mM Tris, pH 7.5, 50 mM NaCl, 1 mM DTT, 1 mM MgCl₂, 4% glycerol, 0.5 mM EDTA, 5 μg/ml poly-dIdC (Sigma P4929)) until the total volume reached 25 μl. After incubating for 30 min at room temperature, the 150-μl Promega buffer (without poly-dIdC) containing 10-μl Ni Sepharose 6 Fast Flow resin (GE Healthcare 17-5318-01) equilibrated in binding buffer was added into the mixture and incubated for 60 min with gentle shaking at room temperature. After binding, the resin beads were consecutively washed with gentle shaking for 12 times with 200 μl of Promega binding buffer (without poly-dIdC). It took 5 min for each washing. Subsequently, the residual moisture on beads was carefully cleared by a soft centrifuging at 500 g for 30 s, and then the bound DNA was re-suspended by using 200 μL of double-distilled water. Finally, about 20 μL bound DNA was subject for 18 cycles of PCR amplification using Phusion DNA polymerase (NEB M0530L) with the primers in Supplementary Table 1c, and the resulting PCR products were used as selection ligands for the next cycle of HT-SELEX. This process was repeated four times. After each cycle, the purified PCR products from each HT-SELEX cycle were pooled and sequenced using Illumina HiSeq Xten or BGI MGISEQ 2000 sequencer.

**HT-SELEX data analysis.** Raw sequencing data were binned according to barcodes for each sample. Sequences from the 40-nt random region without bases annotated as N were used for further analyses. PWM models were generated using initial seeds identified using Autoseed[4,6]. Exact seeds, cycles, and multinomial models are shown in Supplementary Data 4. All motif seqlogos were generated using the R package ggseqlogo[58].

**Network analysis of the similarity between the PWMs.** We calculated the similarities of all pairs of 118 PWMs using SSTAT[4,59] (parameters: 50% GC-content, pseudocount regularization, type I threshold 0.01)[4]. We generated a network containing two types of nodes, one type representing TF-binding profiles, and another type representing TF proteins. TF protein nodes were connected to their binding models, and the binding models were further connected to each other if their SSTAT[4] similarity score (asymptotic covariance) was greater than $1.5 \times 10^{-5}$. Finally, the network was visualized using Cytoscape software v3.7.2[60].

**Electrophoretic mobility shift assay.** DNA probes were PCR-amplified using primers listed in Supplementary Table 1b. The probe (30 ng) was mixed with various amounts of protein in 20 μl of gel shift buffer (10 mM Tris-HCl, pH 7.4, 50 mM KCl, 5 mM MgCl₂, 10% glycerol). After incubation at room temperature for 30 min, the reactions were run by 6% polyacrylamide gel electrophoresis at 100 V for 60 min. The gels were subjected to DNA dye for 5–10 min, and photographed by using the gel imaging system (Bio-Rad). The assay was repeated at least twice with similar results.

**Construction of deletion mutant.** A SacB-based strategy was employed for the construction of gene knockout mutants[61,62]. The pK18mobsacB suicide plasmid[63] was digested by using *Eco*RI and *Hind*III. The upstream (~1500 bp) and downstream (~1000 bp) of the transcriptional regulator open-reading frame were amplified from *P. syringae* 1448A genome. All primers are listed in Supplementary Table 1b. Then, the *Eco*RI and *Bam*HI digested upstream (~1500-bp) and *Bam*HI and *Hind*III digested downstream fragments were ligated together by T4 DNA ligase. Then, the ligated DNA products were inserted into the digested pK18mobsacB plasmids (*Eco*RI and *Hind*III digested) using ClonExpress MultiS One Step Cloning Kit (Vazyme, China) generating the pK18mobsacB-TF plasmid. The successfully constructed plasmids were electroporated into *P. syringae* 1448A strain with selection for kanamycin resistance. Colonies were selected for loss of sucrose susceptibility on KB agar plates containing 5% sucrose. The *P. syringae* 1448A transcriptional regulator deletion mutants were further confirmed by real-time quantitative PCR (RT-qPCR) to detect the mRNA level of themselves.

**Promoter activity detection.** To report the activity of the promoter in bacteria, we cloned and inserted the corresponding promoters sequence to the promoter-less plasmid pMS402 fusing luciferase gene (Supplementary Table 1b). The pMS402 plasmid was digested by using *Bam*HI in advance. The successfully constructed plasmids were electroporated by using MicroPulser (Bio-Rad) with 1.8-kv measurement during every electroporation into mutants and *P. syringae* 1448A wild-type, respectively. After 48 h, the single colony on each plate was picked and cultured at the mid-log growth phase ($OD_{600} = 0.6$). The luminescence value (counts per second, cps) of bacteria was recorded using a 96-well white microplate in Biotek microplate reader with luminescence fiber optics type. The optical density of corresponding bacteria in each well was determined immediately using a 96-well transparent bottom cell culture plate of Fisher Scientific microplate reader at 600 nm. The promoter activity was evaluated by the ratio of luminescence value and $OD_{600}$ value.

**Quantitative RT-qPCR.** For real-time quantitative PCR (RT-qPCR), all strains were cultured at 28 °C, 220 rpm overnight in KB until $OD_{600}$ to 0.6. To harvest the bacteria, the cultures were centrifuged at 6000 ×*g* for 1 min. RNA purification was performed by using RNeasy minikit (Qiagen). RNA concentration was measured by Nanodrop 2000 spectrophotometer (ThermoFisher), and the cDNA synthesis was performed using a FastKing RT Kit (Tiangen Biotech). RT-qPCR was performed by SuperReal Premix Plus (SYBR Green) Kit (Tiangen Biotech) and prepared by following the manufacturer's instruction. Each reaction was performed in triplicate in 20 μl reaction volume with 20 ng cDNA and 16S rRNA of *P. syringae* 1448A as an internal control. For each reaction, 100 nM primers (Supplementary Table 1b) were used for RT-qPCR. The fold change represents the relative expression level of mRNA, which can be estimated by the values of $2^{-(\Delta\Delta Ct)}$, with 16S rRNA as the reference. All the reactions were conducted with two biological repeats.

**Analysis of the transcriptional regulatory networks.** We first scanned the *P. syringae* reference genome with 118 PWMs using FIMO[64], and then used bedtools (v2.25.0)[65] to annotate all putative binding sites. We generated transcriptional regulatory networks for seven important systems of *P. syringae* that contained two types of nodes, one type representing TF proteins, and another type representing targets. TF protein nodes were connected to their targets. All networks were visualized using Cytoscape software v3.7.2[60]. We used the hypergeometric distribution to calculate the statistical significance (*P* value) and thus analyze the master regulators for each specific pathway and visualize it using the python matplotlib package: the probability of having the number of genes targeted by a TF (*n*) belonging to a specific pathway (k) could be described with the hypergeometric distribution, from the genes in that pathway (K) out of the total number of genes in the *P. syringae* genome (N). Our null hypothesis is that the target genes of a TF are not over-represented in a specific pathway. Therefore, we used the hypergeometric distribution to calculate the statistical significance (*P* value) to see whether the observed number of the given TF-targeted genes was enriched in the specific pathway.

**ChIP-seq analysis.** The raw ChIP-seq data were downloaded from GEO (GSE122629) and were previously published by our lab[37]. The data were mapped to *P. syringae* 1448A genome by using bowtie (v1.2.2)[66]. Only the uniquely mapped reads were kept for the subsequent analyses. Binding peaks were identified using MACS software (v2.1.2)[67]. Peaks were annotated by using bedtools (v2.25.0)[65]. The peaks and reads distribution were visualized by using IGV software (v2.4.14)[68].

**RNA-seq analysis**. The raw RNA-Seq data were downloaded from GEO (GSE122629). These data were previously published by our lab[37]. RNA-seq data were mapped to the *P. syringae* 1448A genome by using HISAT (v2.1.0)[69]. Only the uniquely mapped reads were kept for the subsequent analyses. Gene expression was quantified using GFOLD (v1.1.4)[70]. Reads distribution was visualized by using IGV software (v2.4.14)[68].

**Statistical analysis**. Two-tailed Student's *t* tests were performed using Microsoft Office Excel 2010. *$P < 0.05$, **$P < 0.01$, and ***$P < 0.001$ and results represent means ± SD. All experiments were repeated at least twice.

**Reporting summary**. Further information on research design is available in the Nature Research Reporting Summary linked to this article.

## Data availability

Sequencing data have been deposited in Gene Expression Omnibus (GEO) under accession number GSE146697. Other data are available in this article and its supplementary information, or from the corresponding authors upon request. Source data are provided with this paper.

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

## Acknowledgements

This work was supported by the National Natural Science Foundation of China Grants (81873642 to J.Y., 31900443 to W.S., 31670127 and 31870116 to X.D.), the General Research Funds of Hong Kong (21103018 and 11101619 to X.D.; 21100420 to J.Y.), the City University of Hong Kong (7200595, 9667188, 9610424, and 7005314 to J.Y.), Opening Foundation of Key Laboratory of Resource Biology and Biotechnology in Western China (Northwest University), the Ministry of Education of PRC, and the China Postdoctoral Science Foundation (2019M663794 to L.F. and 2019M663799 to W.S.), Innovation Technology Funds of Hong Kong (ITS/195/18 to X.D.).

## Author contributions

X.D., J.Y., L.F., and T.W. conceived the project. F.L., T.W., C.H., X.L., L.G., J.L., X.S., and N.W. carried out experiments. W.S., Y.Y., X.D., and J.Y. performed data analysis. J.Y., X.D., L.F., T.W., W.S., and C.H. wrote the paper.

## Competing interests

The authors declare no competing interests.
