## [Peer Review File · Nature Communications]

REVIEWER COMMENTS

Reviewer #1 (Remarks to the Author):

In this manuscript, the authors use HT-SELEX to identify binding sites for a large proportion of the transcription regulators in the *P. syringae* genome, corroborate some of these findings via previously published ChIP-Seq, and de novo test others using reporter gene expression and DNA binding assays. This study represents a large leap forward in our understanding of transcription regulator binding sites in a single organism and points out a few interesting connections between regulon members. This work will benefit research in *P. syringae* greatly, as well as other Pseudomonads who possess conserved orthologs to many of these transcription regulators. I am enthusiastic about publication of this data and my comments lay almost entirely in the way the manuscript is written and some of the statements made.

Only Major Issue: Text scope exceeds data scope -

The introduction and parts of the discussion are too grandiose considering this study does not address questions of transcription factor evolution or evolution of binding sites. The readers would be much better served if the authors stick to introducing our understanding of bacterial transcription regulator targets in the introduction. Even in the second paragraph, where the authors finally get to bacterial regulation, they begin much too broadly, as their analysis is really only specifically informative for a small group of the gamma-proteobacteria (or whomever picked these regulators up via HGT) since transcription factor binding site evolution is so much more rapid than coding sequence evolution. The authors even admit the limitations of their study to predict beyond the Pseudomonads in line 87. So, to reiterate, skip the grandiosity and introduce readers to the very specific topic covered here.

Minor notes:

Line 60: "transcription code" or "regulatory code", not "genetic code" (even if you put it in quotes), as that was already taken about 60 years ago.

Line 64-65: Unclear of meaning... All genomes are fundamentally evolved, as every extant genome on earth has been evolving for the exact same amount of time (since there was a single ancestral genome). Rephrase.

Line 122: I know the authors are using the TF database, but nearly all microbiologists working on these topics do not group the TFs by the names used in this manuscript. Even if the authors prefer the new family nomenclature, the readers would be helped greatly if they also use the more standard family designations (i.e. TetR family, LysR family, AraC/XylS family, etc) at least upon first use.

Line 136: Many statements in this manuscript are written as if they are novel thoughts. Most I'm not too worried about, but the conclusion of this line (that TFs in the same family have different binding sites) has been known for decades and is actually a critical component and outcome of target differentiation after gene duplication for the regulators.

Line 176: "dismissed" is the wrong word, and while the sentence is not written very clearly, I think the authors are probably trying to say that "we expected absence of PcaQ in a pcaQ deletion strain to result in reduced (or diminished) expression, but interestingly..."

Line 202: This site likely does not represent a trimer, but a dimeric site on each strand in opposite orientations. This is typical of MarR sites in *E. coli* and many other bacteria.

Line 215: This has been known for decades and is the whole way that MarR and its associated transcriptional regulators involved in heavy-metal and antibiotic resistance work - autorepression

of self, induction of other targets. Not a new finding, although good that their method was able to detect this well-described phenomenon.

The last paragraph of the results is very poorly written and not clear even when puzzled through.

Figure 6: The functional annotation of downstream regulation of bacterial TFs is incredibly poor and therefore this is really of no value.

Reviewer #2 (Remarks to the Author):

This paper reports a lot of very useful information, but gives us no new biological insights other than that there is a transcription factor network. The results are convincing and interesting but this type of Selex approach has inbuilt issues when it comes to Transcription factors whose binding is dependent on ligands or covalent modification.

Reviewer #3 (Remarks to the Author):

Fan, et al. report the in vitro DNA-binding specificities of 100 transcription factors (representing 26 DBD families) of the plant-pathogen *Pseudomonas savastanoi* pv. *phaseolicola* 1448A. DNA-binding specificities of purified proteins were measured by HT-SELEX. This set of specificities, summarized as 118 position-weight matrices (PWM), represents 33% of the TF repertoire of this Bacterium. TF specificities were then used to identify putative genome-wide binding sites and the respective gene targets for each TF. Several binding predictions to genomic sequences were confirmed by in vitro binding (EMSA), gene reporter, and ChIP-seq assays. The authors drew networks of virulence-associated pathways connecting TFs to their respective predicted target genes. Target genes for each TF were tested for significant enrichment in virulence-associated pathways to identify 'master regulators' of such pathways.

Large-scale studies characterizing prokaryotic TFs lag, when compared with their eukaryotic counterparts. The type of data provided by Fan et al are a first step towards building genome-wide models of gene regulatory networks. The original results reported by Fan et al. are an important contribution to the field and will be of interest to researchers working with TFs in general, gene regulation in prokaryotes, and plant pathogens. Statistical are appropriate and valid and described in details in the Methods section.

I recommend the manuscript for publication in Nature Communications after the following issues are addressed.

Major

1) Out of the 300 TFs tested, only 100 showed "robust enrichment". It would be helpful if the authors commented more in this. For the fraction that did not enriched, do the authors think it was due to protein purification/concentration/stability issues or are these TFs not sequence-specific?

2) Line 135-137. "Unlike metazoan TFs, the present HT-SELEX data revealed that the *P. syringae* TFs in the same family did not always display the similar binding specificities, suggesting the need for a better classification." The authors should comment on how they arrived at this conclusion.

Were they comparing PWMs?

3) Related to previous comment: I'd love to see a figure clustering DNA-binding specificity (top kmers, PWM) vs TF similarity (DBD). An analysis along the lines of Fig1 in Badis et al Science 2009, Fig1 in Jolma et al Cell 2013, Fig4 in Nitta et al eLife 2015 and Fig3 Narasimhan et al eLife 2015.

4) Clear distinction between putative/predicted and validated genomic binding sites should be made. E.g., line 227, 540, and others.

5) What explains distribution of target genes per TF? Why 6 for (PSPPH_2563) and 1,481 for (PSPPH_3577)? Does it have to do with site size (monomer vs dimer), motif degeneracy, genome composition, other?

6) What is the overlap between the set of 274 putative binding sites RhpR (PSPPH_2004) and the 103 chip-seq significant peaks?

7) The authors define master regulators as "a class of functionally crucial TFs that participated in a pathway or a biological event by regulating multiple downstream genes associated with that event." (line 245-247). The evidence the authors for these classifications relies on statistical enrichment of TFs target genes in specific virulence pathways. However, is there any evidence that these TFs are "functionally crucial"? For example, "TrpI, RhpR, GacA and PSPPH_3618 were shown to act as the master regulators in T3SS (line 249)" Has it been shown these are "functionally crucial" for the pathway?

Minor

1) Line 62: The authors should include Badis et al Mol Cell 2008 citation for yeast TFs.

2) Line 90: How do the authors know that the E. coli study "resulted in suboptimal consensus binding motif." Is there a citation for this statement or is it in lab validation? How are the authors defining "suboptimal" defined?

3) Line 227, 540, and other: Highlight the difference between a predicted/putative binding site vs binding site.

4) Line 298: I cannot find Extended Data Fig. 6.

5) Line 427 – The <http://www.transcriptionfactor.org/> cited as the source of the TFs in P. syringae is not working (May 2020).

6) Line 465 – For reproducibility, can the authors state how many PCR cycles were performed during the SELEX rounds?

Reviewer #4 (Remarks to the Author):

Although prokaryotes appear to have a relatively low complexity as compared to multicellular eukaryotes, their gene regulatory are still highly interconnected with various levels of how gene expression is controlled. Mapping a complete network of gene regulation for any relevant species would offer new possibilities to model and understand how the cells react to internal and

environmental cues. This is especially important in the arms race with emergent drug resistant pathogens affecting humans and their life stocks and crops. This manuscript describes the application of an elegant high throughput solution (HT-SELEX) to uncover large parts of the regulatory network of *Pseudomonas syringae* (now *P. savastanoi*) pv. *Phaseolicola*, which is a model organism for several plant pathogenic species, but also related to the major nosocomial pathogen *P. aeruginosa*. Species of *Pseudomonas* are usually environmentally versatile with a large fraction of the genome dedicated to gene regulation, so this work has the potential to underline the power of HT-SELEX in deciphering these regulatory networks.

The authors state to have purified a comprehensive set of 301 transcription factors, of which they were able to successfully analyse a third (100 TFs). Using the sequence data, 118 binding motifs for these TFs were identified, and interactions between TFs detected. Multiple TFs have been analysed more thoroughly to validate the findings using several independent methods (ChIP, EMSA and qPCR). A special focus was put on the regulation of virulence factor pathways and several novel connections were found.

The manuscript is well written and the presented results are convincing and well presented. Thus, it has the potential as a blueprint for similar studies in other organisms, for which a thorough analysis of the gene regulatory network is required.

One major issue however caught my attention and this concerns a very central aspect of the study, so it is absolutely necessary to clarify this: according to the description, the TFs have been purified by cloning and expressing their respective genes in the pET28a vector and recovering the TF proteins using the His-6-tag provided by the vector. The primers used for this are listed in Suppl. Table 1b and there are two apparent problems with these primers:

- 1) Both forward and reverse primers were designed with a BamHI restriction site. How did the authors make sure that the genes were inserted into the plasmid in the correct orientation? I would expect that the genes have to be in the orientation as the promoter present on the plasmid (T7) and the His6-locus to be able to purify the actual protein.
- 2) The reverse primers without exception all still include the STOP-codons of the respective genes. This should prevent attachment of the His6-Tag and thus also purification of the protein. Maybe there is a mistake in the method description or I got something completely wrong but I don't understand how it was possible to successfully purify 100 transcription factors with the described method. It is absolutely necessary that this is corrected and explained, because if I got things right, it shouldn't be possible to reproduce the results with the methods as described.

Additional minor comments:

L. 102: "assigning 25 TFs targeting genes related to virulence"

L. 159: how is a tail-to-tail orientation defined and how is it different from head-to-head? How do you determine directionality of the motif?

L. 196: "site immediately"

L. 338ff: this section is difficult to understand. What exactly is unlikely and how is this indicating ancient evolution?

L. 348: what are "biological events of these creatures"? Also, the term "creatures" is very fitting for microbes.

L. 349: how is the number of bacteria affecting "our daily life" increasing? Do you mean that our knowledge on this connection is increasing, the number of bacteria or the number of species?

L 374f: "suboptimal quality of PWM" due to limited number of actual binding sites – maybe true, but on the other hand, ChIP-detected sites are evolved sites while HT-SELEX derived PWM are somewhat artificial. There is a chance that native binding sites have additional features that are not uncovered by this method (e.g. DNA modifications, context of promoter site including sigma-factor binding sites, etc.).

L. 381: The conclusion "evolutionarily preferred" only holds for humans (or eukaryotes) but not for bacteria. Thus, it seems reasonable to assume that both modes have their advantages, depending on the context (prokaryote or eukaryote).

L. 382: "is by dramatically higher than..." What are you trying to say? I didn't get the point...

L. 427: "were identified"

L. 495: "Construction of deletion mutant"

L. 498: "open reading frame"

L. 536: Which gene was used as a reference to calculate delta-delta-Ct?

L. 544: Which hypothesis was used for testing with the hypergeometric distribution?

Fig 1 (L. 826): By "models" you mean PWMs?

Fig 2 (L. 843): This information (replicates etc.) is redundant in the figure description and could be mentioned only once for the whole figure.

Fig 3D: the sequence logos are inconsistently presented, the ChIP-logo seems to use a different scale, not adding up to the same height.

A RESPONSE LETTER TO THE REVIEWER COMMENTS

We greatly appreciate all referees' thoughtful comments and their very constructive advices. While all reviewers provided positive remarks, each referee also raised a number of concerns respectively that we have now fully addressed in the revised manuscript, primarily including: (1) addition of new experimental data to further demonstrate the biological insights of our works, (2) making modifications of some inappropriate statements and (3) extended clarification of experimental details that had caused confusion in the original submission.

A high-level summary of the reviewers' critiques, along with key changes made to address them, is outlined below. The detailed point-by-point responses to all comments are provided thereafter: the reviewers' original critiques are directly cited in *Italic* font followed by our response in Roman typeface. We sincerely thank the four referees for their great input that helped us make significant improvements in this revised manuscript.

(1) New experimental data to demonstrate the biological insights

We have generated a deletion strain for the TF PSPPH_3618, which we identified as a master TF in the virulent T3SS pathways. Our RT-qPCR results demonstrated that the transcriptional levels of its target genes (*avrB2* and *rhpP*) were significantly lowered in the Δ PSPPH_3618 strain than the wild-type strain in both KB (nutrient-rich) and MM (nutrient-deficient) media (Fig. 3e-h). The plant-infection assay showed that the Δ PSPPH_3618 strain displayed milder disease symptom and >tenfold less bacterial growth than the wild-type strain (Fig. 3i, j). We concluded that PSPPH_3618 was functionally crucial in T3SS-mediated virulence.

(2) Modifications of inappropriate statements

Pointed out by several reviewers that inappropriate statements were made in some places of the original manuscript, we have now revised the statements and focused on the specific aim of our study, providing a resource of TF binding specificity that has helped unravel transcriptional regulation networks and identify key regulators of a variety of functional pathways in *P. syringae*. We removed the grandiose introduction and discussion over TF and genome evolution and get more focused on the scientific questions that our study has answered. We also avoided irrelevant comparison of *E. coli* database with our *P. syringae* dataset in the introduction as they aimed to solve different problems. Other modifications of improper statements are indicated below in direct responses to the reviewers' comments.

(3) Clarification of experimental details

Clear description of method detail was lacking in our first submission which caused confusion to reviewers. We have now modified the text by adding more details, e.g. our high throughput cloning strategy is now described in a step-by-step manner. We have also included a figure to clarify the cloning steps in the response to the reviewer.

Reviewer #1 (Remarks to the Author):

In this manuscript, the authors use HT-SELEX to identify binding sites for a large proportion of the transcription regulators in the *P. syringae* genome, corroborate some of these findings via previously published ChIP-Seq, and de novo test others using reporter gene expression and DNA binding assays. This study represents a large leap forward in our understanding of transcription regulator binding sites in a single organism and points out a few interesting connections between regulon members. This work will benefit research in *P. syringae* greatly, as well as other Pseudomonads who possess conserved orthologs to many of these transcription regulators. I am enthusiastic about publication of this data and my comments lay almost entirely in the way the manuscript is written and some of the statements made.

Response:

We thank the reviewer's generally positive comments on our manuscript and fully agree that the data generated by this work would greatly benefit research in *P. syringae* and other relevant organisms as also supported by other reviewers.

Only Major Issue: Text scope exceeds data scope -

The introduction and parts of the discussion are too grandiose considering this study does not address questions of transcription factor evolution or evolution of binding sites. The readers would be much better served if the authors stick to introducing our understanding of bacterial transcription regulator targets in the introduction. Even in the second paragraph, where the authors finally get to bacterial regulation, they begin much too broadly, as their analysis is really only specifically informative for a small group of the gamma-proteobacteria (or whomever picked these regulators up via HGT) since transcription factor binding site evolution is so much more rapid than coding sequence evolution. The authors even admit the limitations of their study to predict beyond the Pseudomonads in line 87. So, to reiterate, skip the grandiosity and introduce readers to the very specific topic covered here.

Response:

Thank you for the constructive comment. We agree that the previous Introduction and Discussion have unnecessarily gone beyond the central question that this work has aimed to address. In line with the reviewer's advice, we removed the statements which were not addressed in the study in the second paragraph of Introduction part. For example, the irrelevant and improper sentences: "The mutations and binding sites of TFs underlie many human health issues. Over 90% of genetic variants that are associated with various diseases are non-protein coding and heavily enriched in TF binding sites, suggesting the importance of precise binding of TFs in vivo."; "Prokaryotes are extremely ancient organisms that arose during the Precambrian Period 3.5 to 3.8 billion years ago and can survive in every type of environment on this planet, from extremely cold to very hot, even in super saline or very acidic. While cohabiting in the environment with other higher species, some of these microorganisms confer significant impact on their daily life." and "To our best knowledge, the closest species with an available collection of genome-wide TF binding dataset so far is *Escherichia coli*, albeit the two bacteria share very few conserved TFs, lending little help for our study of *P. syringae*. In addition, due to the different design of focus, the *E. coli* study utilized very limited number of DNA sequences in the selection pool resulted in suboptimal consensus binding motif."

Besides, we also removed the following sentences in the Discussion section: “According to a recent estimate, there are around one trillion species of microbes living on earth, far beyond any other species in numbers and diversity. However, the cellular characteristics and function of the vast majority of these microorganisms are underexplored, even though we have started realizing their strong impact on our daily life.”

We hope that the revised manuscript neatly conveys the key information of what the current study primarily focuses on, with a much improved logic flow.

Minor notes:

Line 60: “transcription code” or “regulatory code”, not “genetic code” (even if you put it in quotes), as that was already taken about 60 years ago.

Response:

Thanks! We have changed the “genetic code” to “regulatory code” and paid more attention to the appropriate use of scientific terms throughout the manuscript. Please see Line 50 in the revised manuscript.

Line 64-65: Unclear of meaning... All genomes are fundamentally evolved, as every extant genome on earth has been evolving for the exact same amount of time (since there was a single ancestral genome). Rephrase.

Response:

We apologize for the unclear statement. In this part, we aimed to claim that some structural DBDs already existed in prokaryotic organisms and the sequence preferences of these TFs stay identical along evolution. What keeps changing is the *in vivo* genomic binding sites as these noncoding regulatory elements are evolved faster than the coding sequences of TFs, which has been generally agreed¹. As discussed above, we have decided to remove this part of TF and TFBS evolution of the genome in the Introduction as it is irrelevant to the problem that the current study aims to solve.

Line 122: I know the authors are using the TF database, but nearly all microbiologists working on these topics do not group the TFs by the names used in this manuscript. Even if the authors prefer the new family nomenclature, the readers would be helped greatly if they also use the more standard family designations (i.e. TetR family, LysR family, AraC/XylS family, etc) at least upon first use.

Response:

Thank you for the advice. We have now changed the family nomenclature of the 301 TFs of *P. syringae* to more standard designations as suggested (see revised Supplementary Table 1a and Extended Data Fig. 2a), hoping that the revision could make the manuscript more accessible to the scientists in the relevant microbiology field. In reflection in the text, we changed the following sentences:

- “The majority of TFs (163 TFs) belonged to five families, including LysR family, TetR family, GntR family, OmpR family and AraC/XylS family” (Line 98-99 in the revised manuscript)

- “Of the top five largest TF families, the GntR and OmpR families reached the highest success rate of 50%, while the AraC family had a low success rate of 5%” (Line 107-108 in the revised manuscript).

Line 136: Many statements in this manuscript are written as if they are novel thoughts. Most I'm not too worried about, but the conclusion of this line (that TFs in the same family have different binding sites) has been known for decades and is actually a critical component and outcome of target differentiation after gene duplication for the regulators.

Response:

We fully agree with this reviewer, and we are sorry for the inappropriate statements. In earlier studies with metazoan TFs, we observed that TFs from the same DBD structural families tended to bind similar sequences and the similarity of amino acid sequence helps predict TF DNA-binding specificity^{2,3}. We also appreciate that diversity of DNA specificity was widely detected in TFs within the same structural family, in particular for zinc finger factors^{2,3}.

In Line 136 of our original submission, we compared the clustering analysis of the *P. syringae* motifs (Fig. 1b and Response Figure 1) with that in human and other metazoan species, and found the poor consistency between the family classification and motif similarity of TFs in *P. syringae*. For example, within LysR family, the binding motifs for TrpI, PSPPH_1259, PSPPH_3022, PSPPH_3079, PSPPH_2469, PSPPH_3611, PSPPH_2921, and PcaQ varied from each other (Please see Part 1 in Response Figure 1). This is most likely due to the fact that the family classification which we referred to was not fully based on the similarity of DBD structures. Some TFs of the LysR family probably could be classified into several subclasses based on the presence or absence of other domains. Unlike higher species, e.g. *Drosophila*, mouse, and human, for which sophisticated investigations had been conducted on TFs to carefully characterize and classify them into well-defined structural categories, e.g. homeodomain, T-box, forkhead, etc, TFs of *P. syringae* were roughly sorted into multiple functionally related families, e.g. LysR, TetR, and GntR, as well as a few Helix-Turn-Helix (HTH) large families. In addition, we found that the amino acid sequence similarity of TFs within the same family was not significantly higher than that between different families. For instance, PilR, PSPPH_0146, NtrC, PSPPH_4448, FleR, PSPPH_3907, and TsiR were sorted into the “Fis family” (marked in red in Response Figure 1), while the phylogenetic distance of their DBD amino-acid sequence is far apart (Please see Part 2 in Response Figure 1).

Therefore, to avoid confusion, we have now removed the comparison of DNA binding specificity of TFs in different families and rephrased the manuscript. Please see Line 111.

Response Figure 1 | The binding sequences comparison among 100 TFs from different families. Phylogenetic tree analysis revealed the identities of their DBD amino-acid sequences. Analyses were performed with the MEGA7 software⁴. One hundred of DBD amino-acid sequences were “Aligned By Muscle”. The Phylogenetic tree was constructed by “Neighbor-Joining” and the evolutionary distances were computed by the p-distance. Part 1 shows an example of different binding sequences happening in TFs which belong to LysR family. Part 2 compared the phylogenetic relationship between identities of their DBD amino-acid sequences among different TFs in Fis family.

Line 176: “dismissed” is the wrong word, and while the sentence is not written very clearly, I think the authors are probably trying to say that “we expected absence of PcaQ in a pcaQ deletion strain to result in reduced (or diminished) expression, but interestingly...”

Response:

Yes, we have now changed the confusing word “dismissed”. The new text reads: “We then constructed a pcaQ gene deletion strain (Extended Data Fig. 2c), and expected that the absence of PcaQ would result in dysregulation of its target genes.” Please see Line 152-154.

Line 202: This site likely does not represent a trimer, but a dimeric site on each strand in opposite orientations. This is typical of MarR sites in E. coli and many other bacteria.

Response:

We appreciate the reviewer for pointing out this misinformation in our original submission. Indeed, MarR binds in a head-to-head homodimeric manner. The monomeric half site begins with a “TT” dinucleotide (**Response Figure 2**). We have now corrected our classification in the manuscript and corresponding figures. Please see Line 127, 128 and 136 in manuscript and Fig. 2b, 2e.

Response Figure 2 | PWM logo of PSPPH_3297
The blue arrow indicates the monomeric half site showing that PSPPH_3297 binds in a head-to-head homodimeric manner.

Line 215: This has been known for decades and is the whole way that MarR and its associated transcriptional regulators involved in heavy-metal and antibiotic resistance work - autorepression of self, induction of other targets. Not a new finding, although good that their method was able to detect this well-described phenomenon.

Response:

We apologize for the inappropriate claim of a novel finding about the uncharacterized TF PSPPH_3297, which belongs to the MarR family. We have now cited a few original findings about MarR in E. coli⁵⁻⁷ and used these examples to validate our finding, which is consistent with a well-established notion. Please see Line 181-183 for update in the revised manuscript.

The last paragraph of the results is very poorly written and not clear even when puzzled through.

Figure 6: The functional annotation of downstream regulation of bacterial TFs is incredibly poor and therefore this is really of no value.

Response:

We apologize for the confusing writing of the last paragraph and the mistake of the relevant figure number. We thank the reviewer for pointing out the unnecessary of functional annotation of TFs and Fig. 5 (GO annotation). To avoid further puzzling readers, we have now removed previous Fig. 5 and the last paragraph of the results, since the functional annotations were already discussed in some earlier figures (Figs. 3, 4).

Reviewer #2 (Remarks to the Author):

This paper reports a lot of very useful information, but gives us no new biological insights other than that there is a transcription factor network. The results are convincing and interesting but this type of Selex approach has inbuilt issues when it comes to Transcription factors whose binding is dependent on ligands or covalent modification.

Response:

First, we thank this reviewer for agreeing that “the paper reports a lot of very useful information” and “the results are convincing and interesting”, while we also believe that our results conceptually advance the understanding of transcriptional regulation in many important biological processes in *P. syringae* cells and identified a variety of key regulators in them. In conceiving this work, we aimed to systematically delineate the DNA binding specificities for TFs in *P. syringae* where such large-scale studies characterizing prokaryotic TFs remain lagging. By contrast, similar works for many eukaryotic organisms have already been completed, to our knowledge covering TFs in yeast, nematode, fruit fly, mouse and human. Therefore, such high throughput studies would greatly benefit research in relevant field, which is also agreed by other reviewers.

HT-SELEX is a well-established pipeline to study DNA binding specificities of TFs and has been employed to systematically profile the TF binding modes in fruit fly and human, which involved major contributions from several authors of the current work. The datasets generated by HT-SELEX are generally in high quality and have been well cited by the scientific community. We admit that HT-SELEX approach has some limitations, including the one this reviewer specifically mentioned about the covalent modification, which can be overcome by an *in vivo* methodology such as ChIP-seq. However, ChIP-seq has its own limitations, e.g. it is heavily affected by protein-protein interaction-mediated DNA binding; it also depends on the formaldehyde-dependent crosslinking efficiency. Therefore, these methods could mutually complement each other and provide different information in comprehensively studying TF-DNA binding.

In addition to the dataset itself, we also identified a few master TF regulators in multiple virulence-relevant pathways. For example, we identified four master TFs (RhpR, GacA, PSPPH_3618, and TrpI) as the master T3SS TFs. Among them, RhpR and GacA have been well established as master regulators of T3SS in *P. syringae*⁸⁻¹³. We have also identified that RhpR bound to the promoter of *avrB2* and *rhpR*, confirmed by ChIP-seq (Extended Data Fig. 4b, 4c). The specific RhpR-binding site carries an inverted repeat (IR) element (GTATC-N6-GATAC)⁸⁻¹¹, which is consistent with our HT-SELEX result (Extended Data Fig. 4d). Meanwhile, we performed a series of additional experiments to characterize the biological function of TF PSPPH_3618 in regulating T3SS (Fig. 3d-j):

1) We first cloned the genomic fragments carrying the motif and performed EMSA. The results of EMSA confirmed that PSPPH_3618 interacted with the promoters of both *rhpPC* operon and *avrB2* gene (Fig. 3d).

2) We then generated a deletion strain of the PSPPH_3618 gene (Extended Data Fig. 4d), and detected the transcriptional levels of its target genes (*avrB2* and *rhpP*) in nutrient-rich (KB) and nutrient-deficient medium (MM), respectively (Fig. 3e-h). T3SS genes are expressed at a very low level when grown in KB, but induced to high levels in MM or in plants^{14,15}. The *rhpPC* operon regulates the T3SS¹⁶, while *avrB2* encodes the T3SS effector AvrB2 and enhances

bacterial virulence^{15,17}. Our RT-qPCR results demonstrated that the transcriptional levels of *avrB2* and *rhpP* were significantly lower in Δ PSPPH_3618 than wild-type strain in both media (Fig. 3e-h).

3) Third, we infiltrated the wild-type and Δ PSPPH_3618 strains into the primary leaves of bean plants. The phenotype was photographed 6 days post-inoculation, which showed that the Δ PSPPH_3618 strain infected sites displayed milder disease symptom and tenfold less bacterial growth than the wild-type strain infected sites on the same leaves (Fig. 3i, j).

In sum, we have demonstrated that the uncharacterized TF PSPPH_3618 was functionally crucial in T3SS pathway and further demonstrated the novel biological insights in the present study. We revised Fig. 3 and Extended Data Fig. 4 and added these results to the manuscript. Please see Line 258-277.

Reviewer #3 (Remarks to the Author):

Fan, et al. report the in vitro DNA-binding specificities of 100 transcription factors (representing 26 DBD families) of the plant-pathogen *Pseudomonas savastanoi* pv. *phaseolicola* 1448A. DNA-binding specificities of purified proteins were measured by HT-SELEX. This set of specificities, summarized as 118 position-weight matrices (PWM), represents 33% of the TF repertoire of this Bacterium. TF specificities were then used to identify putative genome-wide binding sites and the respective gene targets for each TF. Several binding predictions to genomic sequences were confirmed by in vitro binding (EMSA), gene reporter, and ChIP-seq assays. The authors drew networks of virulence-associated pathways connecting TFs to their respective predicted target genes. Target genes for each TF were tested for significant enrichment in virulence-associated pathways to identify 'master regulators' of such pathways.

Large-scale studies characterizing prokaryotic TFs lag, when compared with their eukaryotic counterparts. The type of data provided by Fan et al are a first step towards building genome-wide models of gene regulatory networks. The original results reported by Fan et al. are an important contribution to the field and will be of interest to researchers working with TFs in general, gene regulation in prokaryotes, and plant pathogens. Statistical are appropriate and valid and described in details in the Methods section.

I recommend the manuscript for publication in Nature Communications after the following issues are addressed.

Response:

We thank this reviewer's enthusiastic review comments on our work. Indeed, a comprehensive study to characterize the DNA binding specificities of prokaryotic TFs is largely missing. This study mainly aims to fill up such a gap and provide a resource for scientists in the relevant field.

Major

1) Out of the 300 TFs tested, only 100 showed "robust enrichment". It would be helpful if the authors commented more in this. For the fraction that did not enriched, do the authors think it was due to protein purification/concentration/stability issues or are these TFs not sequence-specific?

Response:

We have carefully inspected the protein concentration for each HT-SELEX experiment and tried to use similar amount of proteins across the entire set (100-200 ng). Based on our experience from previous similar works, the amount of TF proteins used in this work ought to be sufficient for sequence enrichment to generate a motif if any. To exclude the technical issue, we also carried out replicative experiments for some TFs where the expression was relatively high, but consistently all of them turned out still unsuccessful in reaching a specific motif. Therefore, we feel that the failure of sequence enrichment of some TFs is most likely due to the fact that they are not sequence specific TFs or the sequence preference of them is generally too weak in relative to other successful cases.

We now add the possible explanation into the Discussion section, which reads: “We also performed replicative experiments for the TFs without enrichment sequence, but consistently all of them turned out still unsuccessful in reaching a specific motif. Therefore, the failure of sequence enrichment of some TFs is likely owing to that they are not sequence specific TFs or the sequence preference of them is generally too weak in relative to other successful cases.” Please see Line 350-354.

2) Line 135-137. “ Unlike metazoan TFs, the present HT-SELEX data revealed that the *P. syringae* TFs in the same family did not always display the similar binding specificities, suggesting the need for a better classification.” The authors should comment on how they arrived at this conclusion. Were they comparing PWMs?

Response:

Yes, when we compared the motif similarity networks (Fig. 1b and Response Figure 1) with the current TF classification by DBD families, we found the general discordance between these two classifications. We already showed that our HT-SELEX data was highly reproducible and supported by existing knowledge, and thus we are confident that most motifs are in high quality. As discussed above in response to Reviewer 1’s comment, we reasoned that such discordance was likely due to the poor family classification of TFs. There are three major issues underlying the discordance between the two categorizations:

1) It is most likely due to the poor definition of DBD in the current database. Only very few of these DBDs had been experimentally validated. In the original design of this study, we tried to use the putative DBDs to perform HT-SELEX, like what we did for human TFs^{2,18}, but almost none of them was able to enrich any sequence motif (data not shown). Then, we had to change to the full-length proteins of the same TFs for HT-SELEX which turned out very successful. Therefore, we felt that the current DBD definition (<https://www.pseudomonas.com/>)¹⁹ was not optimal, at least for a substantial fraction of *P. syringae* TFs.

2) Unlike higher species, e.g. fruit fly, mouse, and human, whose TF families are sophisticatedly curated by their DBD structural categories, the TF families for *P. syringae* are mostly classified very roughly by simply sorting them into functionally related families, e.g. LysR, TetR, and GntR, as well as a few Helix-Turn-Helix (HTH) large families^{20,21} (Supplementary Table 1a). We observed that within some family, e.g. LysR, the binding motifs for TrpI, PSPPH_1259, PSPPH_3022, PSPPH_3079, PSPPH_2469, PSPPH_3611, PSPPH_2921, and PcaQ varied from each other (Please see Part 1 in Response Figure 1). This is most likely due to the fact that the family classification which we referred to does not fully reflect the sequence and structure similarity of DBDs.

3) Consequently, we specifically checked the amino acid sequence similarity of TFs within the same family and found that it was not significantly higher than that of TFs between different families. For instance, PilR, PSPPH_0146, NtrC, PSPPH_4448, FleR, PSPPH_3907, and TsiR were sorted into Fis family (marked in red, please see Part 2 in Response Figure 1), while the phylogenetic distance between them based on identities of their DBD amino-acid sequences (<https://www.pseudomonas.com/>)¹⁹ was far apart, indicating that the current family classification is not as well-defined as higher species.

Clearly, more sophisticated studies in defining the *P. syringae* DBDs or prokaryotic DBDs in general are still required, which is beyond the efforts of this work. Therefore, to avoid confusion, we decided to remove such comparison in our revised manuscript.

3) Related to previous comment: I'd love to see a figure clustering DNA-binding specificity (top kmers, PWM) vs TF similarity (DBD). An analysis along the lines of Fig1 in Badis et al Science 2009, Fig1 in Jolma et al Cell 2013, Fig4 in Nitta et al eLife 2015 and Fig3 Narasimhan et al eLife 2015.

Response:

We have performed the suggested analysis (see Response Figure 1 above for the interest of the reviewer). The clustering of DNA binding specificity differentiated from the TF sequence similarity. According to the discussion above (response to major point 2), such analysis may lead to confusion before more careful DBD classification becomes available and therefore we feel it is more reasonable not to display it in the manuscript, but to show to the reviewers for their interest.

4) Clear distinction between putative/predicted and validated genomic binding sites should be made. E.g., line 227, 540, and others.

Response:

We apologized for the unclear description between predicted and validated binding sites. The binding sites identified by screening PWMs in the genome were called "putative" genomic binding sites, among which confirmed by experiments were defined as "validated" genomic binding sites. We have now clearly differentiated these different types of binding sites in the revised manuscript, and noted the definition in its first presence. Please see Line 33, 148, 160, 166, 174, 183, 203, 208, 210, 283, 300, 331, 358, 552, 610, 618, 894, 908, 920, 926, 927, 934, 967, 971, 980, 981, 988, 991, 993, 995, 1005, 1006, 1010, and 1012.

5) What explains distribution of target genes per TF? Why 6 for (PSPPH_2563) and 1,481 for (PSPPH_3577)? Does it have to do with site size (monomer vs dimer), motif degeneracy, genome composition, other?

Response:

Thanks for bringing up this interesting point. In order to check whether the differential number of genomic binding sites by different TFs resulted from the motif degeneracy or the variety of site size, we plotted the number of sites in the genome against the information content for all motifs (Extended Data Fig. 4a and see also below Response Figure 3). No corresponding trend was observed between the two parameters, suggesting that the distribution of target genes per TF was unlikely caused by the motif degeneracy itself but should be attributable to the *P. syringae* genome composition, which could be driven by evolution. We have now added the discussion to the revised version of the manuscript. Please see Line 209-215.

Response Figure 3 | Scatterplot shows the trend of the number of putative genomic sites of a TF by PWM model (x-axis) along the information content of the model (y-axis). Note that there's no significant correlation between the two variables ($p=0.8712$), suggesting that the differential number of targets of different TFs may not result from the motif complexity or degeneracy but more likely depends on the genomic composition. Red dots highlight the two TFs with the most of least number of putative genomic sites shown in Figure 3a.

6) What is the overlap between the set of 274 putative binding sites RhpR (PSPPH_2004) and the 103 chip-seq significant peaks?

Response:

We have generated a Venn diagram to compare the overlapping binding sites between ChIP-seq and PWM prediction (See **Response Figure 4** below, blue and red) of RhpR. There are 16 peaks shared by the 103 ChIP-seq significant peaks and 274 putative binding sites. This most likely owed to the fact that ChIP-seq significant peaks were substantially affected by protein-protein interactions. Formaldehyde-mediated crosslinking was included in ChIP steps to fix the cells before pulldown. Therefore, the sites where RhpR did not directly bind but mediated by its interacting proteins would also be significantly identified. Therefore, we included the prediction resulted from MEME-generated PWM using the 103 ChIP-seq peak sequences (**Response Figure 4**, yellow). As a result, 33 sites were overlapped between the MEME-generated PWM prediction and the significant ChIP-seq peaks, inferring that only a small fraction of the ChIP-seq peaks was attributed to the direct binding while many of the 103 peaks were likely through protein-protein interactions. In addition, the number of putative binding sites by PWM prediction also heavily depends on the choice of statistic cutoff, which will conceivably affect the overlapping. Similar discordance between PWM prediction and ChIP-seq has been constantly reported and well-known²².

Response Figure 4 | Venn diagram shows the overlapping between the 103 RhpR ChIP-seq peaks (blue) and 274 putative binding sites predicted by the HT-SELEX generated motif (red) or 90 sites predicted by the MEME-generated motif using ChIP-seq peaks.

7) The authors define master regulators as “ a class of functionally crucial TFs that participated in a pathway or a biological event by regulating multiple downstream genes associated with that event.” (line 245-247). The evidence the authors for these classifications relies on statistical enrichment of TFs target genes in specific virulence pathways. However, is there any evidence that these TFs are “ functionally crucial” ? For example, “ TrpI, RhpR, GacA and PSPPH_3618 were shown to act as the master regulators in T3SS (line 249)” Has it been shown these are “ functionally crucial” for the pathway?

Response:

We thank the reviewer for pointing out this important issue. We have now added some existing references by citing others' works to support the “functionally crucial” role of TFs in corresponding biological events. Among the four T3SS master regulators, RhpR and GacA have been well studied as T3SS master regulators of *P. syringae*⁸⁻¹³. Meanwhile, we performed a series of additional experiments on PSPPH_3618, which is currently an uncharacterized gene, to further explore its regulatory role in T3SS:

➤ RhpRS

P. syringae invades host plants through a type III secretion system (T3SS), which is strictly regulated by a two-component system (TCS) called RhpRS. RhpRS coordinates the T3SS gene expression depending on the phosphorylation state of RhpR under different environmental conditions. RhpR is a repressor in self-regulation and regulation of T3SS cascade genes to influence bacterial virulence. RhpS functions as a kinase and a phosphatase on RhpR, the phosphorylation state of which can switch its function between virulence and metabolism. Nutrient-rich conditions allows RhpR to directly regulate multiple metabolic pathways of *P. syringae* and the phosphorylation state enables RhpR to specifically control virulence and the cell envelope. Besides, phosphorylated RhpR directly and negatively regulates the T3SS (via *hrpR* and *hopR1*), swimming motility (via *flhA*), c-di-GMP levels (via PSPPH_2590), and biofilm formation (via *algD*), while it positively regulates twitching motility (via *fimA*) and lipopolysaccharide production (via PSPPH_2653)⁸⁻¹¹. The Δ *rhpS* mutant strain has reduced bacterial pathogenicity and does not confer a significant disease symptom in the leaves compared with wild type bacteria, while Δ *rhpRS* double mutant has a severe symptom in the leaves, showing that RhpR is essential for repressing bacterial virulence¹⁰.

➤ GacA

The response regulator of the TCS GacAS, is a known regulator in T3SS of *P. syringae* by directly binding to and activating the expression of T3SS regulators HrpRS, thus modulating the expression of the T3SS cascade genes, including alternate sigma factors *hrpL*^{12,13}. GacA deficiency results in reduced levels of transcripts of several HrpL-independent genes, causing drastic changes in bacterial virulence towards *Arabidopsis thaliana* and tomato^{12,13}. Besides, GacA positively controls the production of regulatory RNAs, *rsmB* and *rsmZ*, the quorum sensing signal by increasing the expression of *ahlI* and *ahlR*, coronatine biosynthetic genes *corR*, and virulence-related gene *salA*^{12,13}.

We have now added the published references of these two genes in regulating T3SS and virulence in the revised manuscript to support our finding, which reads: “RhpR and GacA have been well recognized as T3SS regulators in *P. syringae*⁸⁻¹³. RhpR is a repressor in self-regulation and regulation of a cascade of T3SS genes to influence bacterial virulence, including *hrpR*, *hopR1*, *flhA* and so on⁸⁻¹¹. GacA was found to regulate T3SS by directly binding to and

activating the expression of T3SS regulators HrpRS, and thus modulated the expression of the T3SS cascade genes, including alternate sigma factors *hrpL*^{12,13}.” Please see Line 236-241.

➤ PSPPH_3618

To further verify our findings of newly identified master TF and demonstrate its biological function in T3SS and virulence, we performed additional experiments on the putatively identified master regulator PSPPH_3618, which had been an uncharacterized gene before our study (Fig. 3):

1) We first cloned the genomic fragments carrying the motif and performed EMSA. The results of EMSA confirmed that PSPPH_3618 interacted with the promoters of both *rhpPC* operon and *avrB2* gene (Fig. 3d).

2) We then generated a deletion strain of the PSPPH_3618 gene (Extended Data Fig. 4d), and detected the transcriptional levels of its target genes (*avrB2* and *rhpP*) in nutrient-rich (KB) and nutrient-deficient medium (MM), respectively (Fig. 3e-h). T3SS genes are expressed at a very low level when grown in KB, but induced to high levels in MM or in plants^{14,15}. The *rhpPC* operon regulates the T3SS¹⁶, while *avrB2* encodes the T3SS effector AvrB2 and enhances bacterial virulence^{15,17}. Our RT-qPCR results demonstrated that the transcriptional levels of *avrB2* and *rhpP* were significantly lower in Δ PSPPH_3618 than wild-type strain in both media (Fig. 3e-h).

3) Third, we infiltrated the wild-type and Δ PSPPH_3618 strains into the primary leaves of bean plants. The phenotype was photographed 6 days post-inoculation, which showed that the Δ PSPPH_3618 strain infected sites displayed milder disease symptom and tenfold less bacterial growth than the wild-type strain infected sites on the same leaves (Fig. 3i, j).

In sum, we have demonstrated that the uncharacterized TF PSPPH_3618 was functionally crucial in T3SS pathway and further demonstrated the novel biological insights in the present study. We revised Fig. 3 and Extended Data Fig. 4 and added these results to the manuscript. Please see Line 258-277.

Minor

1) Line 62: The authors should include Badis et al Mol Cell 2008 citation for yeast TFs.

Response:

Thanks for pointing out the missing citation of a blueprint work. We have now cited the Badis et al. Mol Cell (2008) paper. Please see Line 52.

2) Line 90: How do the authors know that the E. coli study “resulted in suboptimal consensus binding motif.” Is there a citation for this statement or is it in lab validation? How are the authors defining “suboptimal” defined?

Response:

We thank this reviewer for reminding us of the inappropriate statement about the E. coli work, which we fully appreciated as an invaluable resource. We initially meant that the E.coli motifs by Ishihama et al. (2016)²³ were generated with Genomic-SELEX followed by a chip-

based sequencing identification system, surveying much a smaller number of sequences in TF binding compared with our HT-SELEX: the starting library of Genomic-SELEX, seeded from *E. coli* genomic DNA, had much lower complexity compared with our HT-SELEX using synthesized random DNA; in addition, the chip-based sequence identification only allowed a coverage of “43,450 species of 60-base probes spaced at an average interval of 45 bp along the *E. coli* genome”²³. We appreciated that these *E. coli* genomic libraries had advantages over random DNA in that the k-mer composition was more similar to the *E. coli* genome. For a purpose to identify regulatory targets of uncharacterized TFs in *E. coli*²³, genomic-SELEX worked very well. However, this was also likely to introduce bias as these *E. coli* motifs might not fully describe the biochemical affinity of the *E. coli* TFs when compared to motifs of their orthologues in other species. Therefore, the results may “lend little help for our study of *P. syringae*”. To avoid confusion and more focused on our own results, we removed this part from the Introduction.

3) Line 227, 540, and other: Highlight the difference between a predicted/putative binding site vs binding site.

Response:

We apologized for the unclear description between predicted and validated binding sites. The binding sites identified by screening PWMs in the genome were called “putative” genomic binding sites, among which confirmed by experiments were defined as “validated” genomic binding sites. We have now clearly differentiated these different types of binding sites in the revised manuscript, and noted the definition in its first presence. Please see Line 33, 148, 160, 166, 174, 183, 203, 208, 210, 283, 300, 331, 358, 552, 610, 618, 894, 908, 920, 926, 927, 934, 967, 971, 980, 981, 988, 991, 993, 995, 1005, 1006, 1010, and 1012.

4) Line 298: I cannot find Extended Data Fig. 6.

Response:

We apologize for the mis-citation of the display item. It should be “Supplemental Data 3c” instead. We have now corrected it in the revised manuscript. Please see Line 312.

5) Line 427 – The <http://www.transcriptionfactor.org/> cited as the source of the TFs in *P. syringae* is not working (May 2020).

Response:

We also noticed unstable connection from May 2020. This was likely caused by the updating of the database. To avoid confusion, we now refer the text to the original publication, which reads: “The 301 TFs were identified according to Wilson et al²⁴” (see Line 432). We also contacted the authors regarding this issue.

6) Line 465 – For reproducibility, can the authors state how many PCR cycles were performed during the SELEX rounds?

Response:

For each HT-SELEX cycle, 18 PCR cycles were performed to amplify the output DNA library. We have added the information into the **Methods** section. Please see Line 478.

Reviewer #4 (Remarks to the Author):

Although prokaryotes appear to have a relatively low complexity as compared to multicellular eukaryotes, their gene regulatory are still highly interconnected with various levels of how gene expression is controlled. Mapping a complete network of gene regulation for any relevant species would offer new possibilities to model and understand how the cells react to internal and environmental cues. This is especially important in the arms race with emergent drug resistant pathogens affecting humans and their life stocks and crops. This manuscript describes the application of an elegant high throughput solution (HT-SELEX) to uncover large parts of the regulatory network of *Pseudomonas syringae* (now *P. savastanoi*) pv. *Phaseolicola*, which is a model organism for several plant pathogenic species, but also related to the major nosocomial pathogen *P. aeruginosa*. Species of *Pseudomonas* are usually environmentally versatile with a large fraction of the genome dedicated to gene regulation, so this work has the potential to underline the power of HT-SELEX in deciphering these regulatory networks.

The authors state to have purified a comprehensive set of 301 transcription factors, of which they were able to successfully analyse a third (100 TFs). Using the sequence data, 118 binding motifs for these TFs were identified, and interactions between TFs detected. Multiple TFs have been analysed more thoroughly to validate the findings using several independent methods (ChIP, EMSA and qPCR). A special focus was put on the regulation of virulence factor pathways and several novel connections were found.

The manuscript is well written and the presented results are convincing and well presented. Thus, it has the potential as a blueprint for similar studies in other organisms, for which a thorough analysis of the gene regulatory network is required.

Response:

We thank this reviewer for his/her encouraging comments on our current works and we fully agree that the systematic gene regulatory network analysis could raise broad interest and support many relevant studies. Similar works could be generally conducted for other organisms.

One major issue however caught my attention and this concerns a very central aspect of the study, so it is absolutely necessary to clarify this: according to the description, the TFs have been purified by cloning and expressing their respective genes in the pET28a vector and recovering the TF proteins using the His-6-tag provided by the vector. The primers used for this are listed in Suppl. Table 1b and there are two apparent problems with these primers:

1) Both forward and reverse primers were designed with a BamHI restriction site. How did the authors make sure that the genes were inserted into the plasmid in the correct orientation? I would expect that the genes have to be in the orientation as the promoter present on the plasmid (T7) and the His6-locus to be able to purify the actual protein.

Response:

We apologize for lack of clear description of the cloning method. Instead of the regular cloning strategy using restriction enzyme digestion followed by sticky-end ligation, we employed a homologous recombination technology²⁵ using the Vazyme ClonExpress II One Step Cloning Kit (Vazyme Biotech, catalog number: C112-01/02) and followed the manufacture's

instruction to clone the TF cDNA into pET28a expression vector, which easily allowed high throughput cloning of hundreds of genes in one design (see **Response Figure 5** below for the mechanism and workflow, which was adapted from the user manual provided by the manufacture).

Response Figure 5 | Cloning strategy,

- BamHI was used to linearize the empty pET28a vector.
- Each forward PCR primer carried a 20-bp sequence identical to the linearized plasmid sequence at the 5'-end of the cutting site followed by the gene-specific sequence.
- Similarly, each reverse PCR primer carried a 20-bp sequence identical to the linearized plasmid sequence at the 3'-end of the cutting site followed by the gene-specific sequence.
- Therefore, the homologous match of the two 20-bp recombination fragments determined the direction of the target gene in the expression vector.

We have now provided more details in the **Methods** section to further clarify the cloning mechanism and strategy. The following text has been added to Line 435-445. A homologous recombination technology using the Vazyme ClonExpress II One Step Cloning Kit (Vazyme Biotech, catalog number: C112-01/02) was employed to clone the TF cDNA into pET28a expression vector:

- Firstly, the BamHI-linearized pET28a vector and individual TF PCR products (containing 20-bp overlapped sequences on 5'- and 3'-end, respectively) were mixed in the molar ratio of 1:2, and then incubated with recombinase (Vazyme ClonExpress II One Step Cloning Kit, Vazyme Biotech) for 30 min at 37 °C.
- Secondly, the recombination products were chemically transformed into *E. coli* (DH5a) competent cells with at 42 °C for 1 min.
- Finally, the successful constructs verified by BamHI digestion were further transformed into the *E. coli* BL21 (DE3) strain protein production.

2) The reverse primers without exception all still include the STOP-codons of the respective genes. This should prevent attachment of the His6-Tag and thus also purification of the protein. Maybe there is a mistake in the method description or I got something completely wrong but I don't understand how it was possible to successfully purify 100 transcription factors with the described method. It is absolutely necessary that this is corrected and explained, because if I got things right, it shouldn't be possible to reproduce the results with the methods as described.

Response:

In the original empty pET-28a plasmid, there are two sets of His-6-tag sequences (see below Response Figure 6 the MCS map of the pET-28a plasmid we used). As we chose to clone our TF gene into the site of BamHI, the N-terminal His-6-tag was retained. To make our result consistent (not to be interfered by the choice of side of His-6-tag), we decided to get rid of the C-terminal His-6-tag by adding a STOP-codon immediately after the TF gene codon regions.

Response Figure 6 | The MCS map of the pET-28a plasmid

Additional minor comments:

L. 102: “ assigning 25 TFs targeting genes related to virulence”

Response:

Thanks. We have rephrased the text accordingly. Please see Line 80-81.

L. 159: how is a tail-to-tail orientation defined and how is it different from head-to-head? How do you determine directionality of the motif?

Response:

In most cases when two monomers of one TF bind in opposite strands of the DNA forming a homodimer, we define the binding mode as the head-to-head orientation. However sometimes, two directions of such opposite homodimers are observed. For example, the monomer motif for TF OxyR is “CTAT”. There are two directions of the homodimer binding sites (see Response Figure 7 below): if the head-to-head orientation is designated “CTATNNNNNATAG”, the tail-to-tail orientation is defined as “ATAGNNNNNNNCTAT”. We hope that this illustration can better clarify both directions of the orientation. To avoid confusion, we clarified the point in the manuscript. Please see Line 132-136.

Response Figure 7| Illustration of homodimer orientation.

Left figure shows both directions of the homodimer orientation. The red arrows above the motif logos indicate the orientation of a monomer half site (CTAT).

L. 196: “ site immediately”

Response:

Thanks. We have corrected it. Please see Line 174-175.

L. 338ff: this section is difficult to understand. What exactly is unlikely and how is this indicating ancient evolution?

Response:

We apologize for the confusion. We meant to state that the convergent function (shared GO terms) was unlikely due to their similarity of binding motifs but by co-binding to adjacent positions in the genome forming dense TF clusters, which are commonly observed in mammalian genomes²⁶. Another reviewer also pointed out that this paragraph was poorly written and the GO analysis was not necessary as it provided little additional information following Fig. 3 and 4. We agree and therefore decided to remove Fig. 5.

L. 348: what are “ biological events of these creatures” ? Also, the term “ creatures” is very fitting for microbes.

Response:

We initially meant that these many bacteria species have not been characterized, in terms of their special biological features and the impacts in the environment. We agree this statement reads a bit odd and contains almost no useful information. In order to directly focus on the results and impact of our study, we have removed the first paragraph of discussion in the original submission.

L. 349: how is the number of bacteria affecting “ our daily life” increasing? Do you mean that our knowledge on this connection is increasing, the number of bacteria or the number of species?

Response:

We apologize for these unclear statements. Here, we meant our knowledge about the connection is increasing as suggested by this reviewer. For example, many human health issues have been notably connected to the gut microbiota²⁷. However, following another reviewer’s comment, we removed this part in the revised manuscript as it is not so relevant to our work.

L 374f: “ suboptimal quality of PWM” due to limited number of actual binding sites –maybe true, but on the other hand, ChIP-detected sites are evolved sites while HT-SELEX derived PWM are somewhat artificial. There is a chance that native binding sites have additional features that are

not uncovered by this method (e.g. DNA modifications, context of promoter site including sigma-factor binding sites, etc.).

Response:

We fully agree that HT-SELEX and ChIP-seq are mutually complementary in solving different questions in terms of TF-DNA interactions. HT-SELEX uses synthesized random sequences as cycle 0 input to derive PWM that primarily describes the pure biochemical preference of DNA sequences for the *E. coli* expressed TFs. It can differ from the ChIP-detected sites in many reasons, including k-mer composition of the starting DNA library, protein modifications or protein-protein interactions existing in the native cells, and DNA modifications, etc. We specifically compared the difference of HT-derived PWM and ChIP-seq-produced PWM, and ChIP-seq peaks, all of which displayed limited overlap (Response Figure 4), suggesting the different information that each method could provide. The original statement was a bit biased towards HT-SELEX-derived PWM models. We have now added more balanced discussion in the revised version of the manuscript, which reads: “We certainly appreciate that ChIP-seq has its own advantage as it detects TF native binding sites that contain additional features e.g. DNA modifications, context of promoter site including sigma-factor binding sites, etc., incapable of being uncovered by *in vitro* studies, like HT-SELEX. Therefore, these methods could mutually complement each other and provide different information in comprehensively studying TF-DNA binding.” Please see Line 374-379.

L. 381: The conclusion “ evolutionarily preferred” only holds for humans (or eukaryotes) but not for bacteria. Thus, it seems reasonable to assume that both modes have their advantages, depending on the context (prokaryote or eukaryote).

Response:

Thanks for the considerate advice. Indeed, the two modes could be advantageous for different organisms. We have now modified the statement in the Discussion. Please see Line 384-385, and 387-392.

L. 382: “ is by dramatically higher than...” What are you trying to say? I didn't get the point...

Response:

Sorry for the grammatical problem. We have now corrected it. Please see Line 385.

L. 427: “ were identified”

L. 495: “ Construction of deletion mutant”

L. 498: “ open reading frame”

Response:

Thanks. We have corrected these mistakes or confusing statements. Please see Lines 432, 508, and 511.

L. 536: Which gene was used as a reference to calculate delta-delta-Ct?

Response:

We apologize for missing the important information. We used the 16s rRNA gene as a reference. We have now revised it to “the fold change represents relative expression level of mRNA, which can be estimated by the values of $2^{-(\Delta\Delta Ct)}$, with 16S rRNA as the reference” in the Methods section. Please see Lines 546-548.

L. 544: Which hypothesis was used for testing with the hypergeometric distribution?

Response:

The probability of having the number of genes targeted by a TF (n) belonging to a specific pathway (k) could be described with the hypergeometric distribution, from the genes in that pathway (K) out of the total number of genes in the *P. syringae* genome (N). Our null hypothesis is that the target genes of a TF are not over-represented in a specific pathway. Therefore, we used the hypergeometric distribution to calculate the statistical significance (p value) to see whether the observed number of the given TF-targeted genes was enriched in the specific pathway. We hope that such explanation could clarify our hypergeometric test and we have added the detail into the Methods section. Please see Line 558-564.

Fig 1 (L. 826): By “models” you mean PWMs?

Response:

Yes, we have added “PWM” preceding the models. Please see Line 880-881.

Fig 2 (L. 843): This information (replicates etc.) is redundant in the figure description and could be mentioned only once for the whole figure.

Response:

We have now checked and revised all the figure legends with similar issues in line with the reviewer’s advice. Thank you.

Fig 3D: the sequence logos are inconsistently presented, the ChIP-logo seems to use a different scale, not adding up to the same height.

Response:

We apologize for the inconsistency when presenting the motifs. The motif seqlogo on the top of Fig. 3d is drawn using R ggseqlogo package based on the PWM from our HT-SELEX experiment, and the bottom one of Fig. 3d is the result of motif enrichment analysis of ChIP-Seq peaks by using MEME suite. We have replaced and adjusted them for the same presenting scale for fair comparison. Please be noted that we re-organized original Fig. 3d to Extended Data Fig. 4d.

References:

- 1 Bird, C. P. *et al.* Fast-evolving noncoding sequences in the human genome. *Genome Biol* **8**, R118, doi:10.1186/gb-2007-8-6-r118 (2007).
- 2 Jolma, A. *et al.* DNA-binding specificities of human transcription factors. *Cell* **152**, 327-339, doi:10.1016/j.cell.2012.12.009 (2013).
- 3 Nitta, K. R. *et al.* Conservation of transcription factor binding specificities across 600 million years of bilateria evolution. *Elife* **4**, doi:10.7554/eLife.04837 (2015).
- 4 Kumar, S., Stecher, G. & Tamura, K. MEGA7: Molecular Evolutionary Genetics Analysis Version 7.0 for Bigger Datasets. *Mol Biol Evol* **33**, 1870-1874, doi:10.1093/molbev/msw054 (2016).
- 5 Duval, V., McMurry, L. M., Foster, K., Head, J. F. & Levy, S. B. Mutational analysis of the multiple-antibiotic resistance regulator MarR reveals a ligand binding pocket at the interface between the dimerization and DNA binding domains. *J Bacteriol* **195**, 3341-3351, doi:10.1128/JB.02224-12 (2013).
- 6 Sulavik, M. C., Gambino, L. F. & Miller, P. F. The MarR repressor of the multiple antibiotic resistance (*mar*) operon in *Escherichia coli*: prototypic member of a family of bacterial regulatory proteins involved in sensing phenolic compounds. *Mol Med* **1**, 436-446 (1995).
- 7 Martin, R. G. & Rosner, J. L. Binding of purified multiple antibiotic-resistance repressor protein (MarR) to *mar* operator sequences. *Proc Natl Acad Sci U S A* **92**, 5456-5460, doi:10.1073/pnas.92.12.5456 (1995).
- 8 Deng, X. *et al.* Molecular mechanisms of two-component system RhpRS regulating type III secretion system in *Pseudomonas syringae*. *Nucleic Acids Res* **42**, 11472-11486, doi:10.1093/nar/gku865 (2014).
- 9 Xie, Y. *et al.* *Pseudomonas savastanoi* Two-Component System RhpRS Switches between Virulence and Metabolism by Tuning Phosphorylation State and Sensing Nutritional Conditions. *mBio* **10**, doi:10.1128/mBio.02838-18 (2019).
- 10 Xiao, Y. *et al.* Two-component sensor RhpS promotes induction of *Pseudomonas syringae* type III secretion system by repressing negative regulator RhpR. *Mol Plant Microbe Interact* **20**, 223-234, doi:10.1094/MPMI-20-3-0223 (2007).
- 11 Deng, X. *et al.* *Pseudomonas syringae* two-component response regulator RhpR regulates promoters carrying an inverted repeat element. *Mol Plant Microbe Interact* **23**, 927-939, doi:10.1094/MPMI-23-7-0927 (2010).
- 12 Chatterjee, A. *et al.* GacA, the response regulator of a two-component system, acts as a master regulator in *Pseudomonas syringae* pv. tomato DC3000 by controlling regulatory RNA, transcriptional activators, and alternate sigma factors. *Mol Plant Microbe Interact* **16**, 1106-1117, doi:10.1094/MPMI.2003.16.12.1106 (2003).
- 13 Cha, J. Y., Lee, D. G., Lee, J. S., Oh, J. I. & Baik, H. S. GacA directly regulates expression of several virulence genes in *Pseudomonas syringae* pv. tabaci 11528. *Biochem Biophys Res Commun* **417**, 665-672, doi:10.1016/j.bbrc.2011.11.124 (2012).
- 14 Huynh, T. V., Dahlbeck, D. & Staskawicz, B. J. Bacterial blight of soybean: regulation of a pathogen gene determining host cultivar specificity. *Science* **245**, 1374-1377, doi:10.1126/science.2781284 (1989).
- 15 Rahme, L. G., Mindrinos, M. N. & Panopoulos, N. J. Plant and environmental sensory signals control the expression of *hrp* genes in *Pseudomonas syringae* pv. *phaseolicola*. *J Bacteriol* **174**, 3499-3507, doi:10.1128/jb.174.11.3499-3507.1992 (1992).
- 16 Li, K. *et al.* Two components of the *rhpPC* operon coordinately regulate the type III secretion system and bacterial fitness in *Pseudomonas savastanoi* pv. *phaseolicola*. *PLoS Pathog* **15**, e1007673, doi:10.1371/journal.ppat.1007673 (2019).

- 17 Selote, D., Shine, M. B., Robin, G. P. & Kachroo, A. Soybean NDR1-like proteins bind pathogen effectors and regulate resistance signaling. *New Phytol* **202**, 485-498, doi:10.1111/nph.12654 (2014).
- 18 Yin, Y. *et al.* Impact of cytosine methylation on DNA binding specificities of human transcription factors. *Science* **356**, doi:10.1126/science.aaj2239 (2017).
- 19 Winsor, G. L. *et al.* Enhanced annotations and features for comparing thousands of *Pseudomonas* genomes in the *Pseudomonas* genome database. *Nucleic Acids Res* **44**, D646-653, doi:10.1093/nar/gkv1227 (2016).
- 20 El-Gebali, S. *et al.* The Pfam protein families database in 2019. *Nucleic Acids Res* **47**, D427-D432, doi:10.1093/nar/gky995 (2019).
- 21 Perez-Rueda, E. & Collado-Vides, J. The repertoire of DNA-binding transcriptional regulators in *Escherichia coli* K-12. *Nucleic Acids Res* **28**, 1838-1847, doi:10.1093/nar/28.8.1838 (2000).
- 22 Gordan, R., Hartemink, A. J. & Bulyk, M. L. Distinguishing direct versus indirect transcription factor-DNA interactions. *Genome Res* **19**, 2090-2100, doi:10.1101/gr.094144.109 (2009).
- 23 Ishihama, A., Shimada, T. & Yamazaki, Y. Transcription profile of *Escherichia coli*: genomic SELEX search for regulatory targets of transcription factors. *Nucleic Acids Res* **44**, 2058-2074, doi:10.1093/nar/gkw051 (2016).
- 24 Wilson, D., Charoensawan, V., Kummerfeld, S. K. & Teichmann, S. A. DBD--taxonomically broad transcription factor predictions: new content and functionality. *Nucleic Acids Res* **36**, D88-92, doi:10.1093/nar/gkm964 (2008).
- 25 Xu, B. *et al.* Molecular and Biochemical Characterization of a Novel Xylanase from *Massilia* sp. RBM26 Isolated from the Feces of *Rhinopithecus bieti*. *J Microbiol Biotechnol* **26**, 9-19, doi:10.4014/jmb.1504.04021 (2016).
- 26 Yan, J. *et al.* Transcription factor binding in human cells occurs in dense clusters formed around cohesin anchor sites. *Cell* **154**, 801-813, doi:10.1016/j.cell.2013.07.034 (2013).
- 27 Shreiner, A. B., Kao, J. Y. & Young, V. B. The gut microbiome in health and in disease. *Curr Opin Gastroenterol* **31**, 69-75, doi:10.1097/MOG.000000000000139 (2015).

REVIEWERS' COMMENTS:

Reviewer #1 (Remarks to the Author):

I thank the authors for considering and appropriately responding to all of my comments.

Reviewer #3 (Remarks to the Author):

The authors have satisfactorily addressed all my concerns and suggestions in the revised manuscript.

Reviewer #4 (Remarks to the Author):

The authors have thoroughly and comprehensively revised the manuscripts. There are no further open questions or apparent issues and therefore I suggest to accept the manuscript for submission.

Best regards,
Andreas Dötsch